# Linguistic structure and language familiarity sharpen phoneme encoding in the brain
Filiz Tezcan [1,2,3] ✉, Sanne Ten Oever [1,2,3], Fan Bai [1], Noémie te Rietmolen[1] & Andrea E. Martin [1,3]

How does the brain turn a physical signal like speech into meaning? It draws on two key sources: linguistic structure (e.g., phonemes, syntax) and statistical regularities from experience. Yet how these jointly shape neural representations of language remains unclear. We used MEG to track phonemic and acoustic encoding during spoken language comprehension in native Dutch, Mandarin-Chinese, and Turkish speakers. Phoneme-level encoding is stronger during sentence comprehension than in word lists, and more robust within words than random syllables. Surprisingly, similar encoding emerges even in an uncomprehended language but only with prior exposure. In contrast, acoustic edges are briefly suppressed early in comprehension. This suggests that the brain's alignment to speech (in phase and power) is robustly tuned by structure and by learned statistical patterns. Our findings show how structured knowledge and experience-based learning interact to shape neural responses to language, offering insight into how the brain processes complex, meaningful signals.

When we know a language, our brain achieves structured meaning (e.g., perceives words, phrases, sentences) from hearing speech or seeing sign. At least two types of information are key in this process: knowledge of linguistic units and how they form structures, and statistical information about the distribution of those units and structures. As such, comprehending language from speech or sign necessitates the conversion, by the brain, of sensory input (e.g., the neural response to the speech or sign envelope) into linguistic units (e.g., phonetic features, phonemes, syllables), which in turn form the structures of language (e.g., morphemes, words, phrases, sentences, discourse). This conversion crucially depends on the brain applying its knowledge of language, across multiple levels of representation, to infer or integrate linguistic units into increasingly abstract structures, but also on what the brain can predict or expect based on its knowledge of language and its experience with it. Much previous research has demonstrated that both distributional knowledge about linguistic units and sequences (e.g., entropy and surprisal) and the presence of linguistic structure (e.g., word onsets, syntactic structure, annotations of syntactic complexity) affect how faithfully neural signals can be reconstructed[1–13]. Neural encoding models can quantify how the neural response aligns with both acoustic and linguistic features of the speech input; this alignment in phase, or phase and power, is often referred to as *neural tracking*[14–17].

In recent years, the investigation of neural tracking in the delta and theta bands, encompassing the alignment of neural activity phases with

acoustic and linguistic features of speech signals, has become an integral component of contemporary theories in speech and language processing. It's been shown that slow neural activity at delta/theta band recorded by MEG/EEG synchronizes with speech stimulus[15,17–24]. Alignment of the phase of the slow neural activity with sensory input is posited as a canonical mechanism influencing temporal aspects of speech perception[25]. Because linguistic information unfolds across timescales spanning both delta and theta ranges, though we do not suggest a strict one-to-one mapping, linguistic representations relevant for theories of neural encoding are likely reflected in modulations across these frequencies. Accordingly, analyzing the broadband signal captures the essential neural responses without imposing biases or assuming an unsubstantiated correspondence between specific linguistic processes and particular frequency bands.

Existing research has demonstrated that neural tracking is modulated by speech intelligibility[16,18,26,27], attention[28–31], language experience[32–35], and also linguistic content of speech[5,11–13,36–40]. However, linguistic features of speech can be correlated with the acoustic features, and for example, the neural tracking of phonemic features can also be explained by the tracking of rapid changes in the acoustic envelope (e.g., acoustic edges[23,41,42]). The acoustic envelope captures slow amplitude changes in speech signals, with its temporal patterns reflecting the timescales of syllables and words. Phoneme onsets correspond to acoustic onsets, termed "acoustic edges", which are derived from half-wave rectified derivatives of the acoustic envelope[1,41].

[1]Language and Computation in Neural Systems Group, Max Planck Institute for Psycholinguistics, Nijmegen, The Netherlands. [2]Department of Cognitive Neuroscience, Faculty of Psychology and Neuroscience, Maastricht University, Maastricht, The Netherlands. [3]Donders Centre for Cognitive Neuroimaging, Radboud University, Nijmegen, The Netherlands. ✉e-mail: tezcanfiliz@gmail.com

To separate neural responses driven by acoustic versus linguistic information at the phoneme level, we incorporated acoustic edge features (an 8 logarithmically spaced frequency band acoustic onset spectrogram, which models sudden changes in the gammatone spectrogram of the audio signal using an acoustic edge detection model) alongside an auditory spectrogram with 8-band that represents the acoustic envelope (Fig. 1A, See "Methods").

We then compared how well these acoustic edge features and phoneme features predicted neural responses. There are only limited number of studies which investigated the dynamic encoding of acoustic and linguistic features concurrently and the effect of comprehension and language proficiency on the neural tracking of acoustic and linguistic features[5,34,35,38,43].

Di Liberto et al.[43] showed that when second language proficiency increases, encoding phonemes in L2 becomes more similar to L1. Tezcan et al.[38] demonstrated that phoneme features, including phoneme onsets, surprisal, and entropy, exhibited enhanced encoding in a comprehended language compared to an uncomprehended one. In the comprehended language, context at the sentence and discourse levels, as operationalized by word entropy values, suppressed the strength of encoding of both acoustic and phoneme features when the predictability of the subsequent word increases. A similar impact of word entropy was found in the uncomprehended language, despite participants not being able to comprehend the language nor the context. However, acoustic features were modulated by word entropy more in the uncomprehended language, whereas phoneme features were modulated more in the comprehended language. One plausible explanation for this modulation is the high correlation between word entropy and word frequency[44], such that the modulation of acoustic and phoneme features may be linked to participants developing sensitivity to acoustic features of high-frequency words that they recognize due to the exposure to the unknown language. Brodbeck et al.[34] showed that when second language proficiency of participants increases, the encoding of acoustic features was suppressed. Similarly, Pérez-Navarro et al.[35] demonstrated that in bilingual children, acoustic features are tracked more strongly for the least experienced language, and semantic information is tracked stronger in the most experienced language. On the other hand, Gillis et al.[5] showed that word surprisal encoded better in Dutch stories compared to a Dutch word list condition and Frisian stories but did not find any difference on the encoding of acoustic features between a native (Dutch) and nonnative (Frisian) language which was partly comprehended. This could be related to the possibility that Frisian language is statistically familiar to Dutch participants, or that it has significant overlap with the statistical properties of Dutch speech acoustics.

These findings suggest that the phase alignment of neural signals with acoustic and linguistic features of speech is dynamically modulated through interactions across multiple levels of linguistic processing during comprehension[45–56]. This modulation reflects both prior language experience and the statistical regularities learned over time[51,52,57]. Within this interactive framework, all subsystems engage in continuous bidirectional communication, rapidly incorporating relevant information from parallel components while simultaneously providing their own computations to other subsystems as they become available. As exposure to a nonnative language increases, linguistic categories such as phonemes become more precisely tuned, reducing reliance on raw acoustic detail because fine-grained sensory information becomes less essential. Likewise, in a native language, higher-level linguistic structures, such as words and sentences, can further refine the tuning of abstract linguistic units across layers of processing[58–60].

Crucially in the above studies, a natural spoken stimulus was used, often an audiobook of short stories featuring discourse contexts where connected speech is supported by previous context and linguistic units can be predicted over a longer timescale than in more controlled experimental designs[61], allowing for rich statistical experience to be built up. A strong test of how linguistic structure influences phoneme and acoustic edge encoding would be an examination of whether higher-level linguistic patterns enhance neural encoding even without naturalistic discourse context. This would reveal how local linguistic structure modulates neural responses to

phonemes and acoustic edges independently of broader conversational meaning. A related question concerns the role of language exposure versus comprehension. Does statistical familiarity with a language's speech patterns, gained through daily exposure without understanding, such as living in a country where one does not speak the local language, affect the neural encoding of acoustic and/or phoneme features in a similar or different way to knowledge of linguistic structure. Batterink and Paller[62,63] have shown that neural tracking of the acoustic envelope in an isochronous artificial speech stream increases by the time of exposure, however it is still not known if exposure to the language affects the tracking of phonemes and acoustic edges in the same way.

In the current study, our objective was firstly to examine the impact of varying levels of language structure (i.e., syllables, words, and sentences) in the absence of predictability of the next word from the discourse context (i.e., words lists or isolated sentences) and to ask if integration of smaller linguistic units into more abstract linguistic units (viz., syllables into words, words into sentences) *suppresses* or *enhances* acoustic and phonemic feature encoding. To achieve this, we contrasted neural responses to sentences and word lists, and words and syllables in Dutch, Chinese and Turkish (Fig. 1B, See "Methods") by analyzing the datasets previously collected from various MEG experiments. Secondly, to disentangle the effects of language familiarity in an uncomprehended language, we compared the neural tracking of acoustic and phonemic features among Dutch- and Mandarin Chinese-speaking participants residing in the Netherlands. They were presented with isochronous two-syllable Dutch words and pseudowords, the latter generated by shuffling the order of syllables. Subsequently, the same participants listened to Mandarin Chinese words and pseudowords. This experimental design allowed us to contrast familiarity with the statistics of a language, as Mandarin-speaking participants were exposed to the Dutch language in daily life but could not comprehend Dutch, while Dutch-speaking participants were not exposed to Mandarin Chinese language in daily life or otherwise, with acoustic and phoneme features carefully controlled.

We applied linear regression models, commonly referred to as temporal response functions (TRFs), to predict neural activity based on acoustic and phoneme features when contrasting words with random syllable streams, and based on acoustic, phoneme, and word-level features when contrasting sentences with word lists (Fig. 1C, see "Methods"). The rationale for this differentiation is that, in the words versus random syllables comparison, the transitional probabilities between words were controlled. In contrast, for the sentence versus word list comparison, these transitional probabilities were not controlled, as the transitional probability between two consecutive words in a sentence is higher in a language. Therefore, we incorporated measures of word surprisal and word entropy, which were computed based on the transitional probabilities of a word given its predecessor using the GPT-2 language model.

We compared the improvement in model accuracy attributed to the inclusion of acoustic and phoneme features, defined as the increase in the proportion of neural signal variance explained by the model when these features were added. This measure reflects how well the real neural signal aligns with the model's predicted signal in terms of both phase and power. In addition, we compared the model weights of the filters learned by the model that represent the neural responses evoked by specific features. This measure enables us to assess differences in the amplitude of neural responses contributing to the predicted signal.

In our earlier study (Tezcan et al.)[38], we found that language comprehension and contextual information modulated the encoding of acoustic edges and phoneme features, with the strongest effects observed in the auditory cortex (AC) and superior temporal gyrus (STG). Other studies also consistently showed that these regions encode acoustic edges and phoneme features[1,4,7,41,64–67]. Additionally, our previous study showed that averaged responses to all phoneme features (onset, surprisal, and entropy) were more consistent across story segments than individual feature responses. This likely reflects the complementary nature of these features: phoneme onset captures time-invariant categorization responses, while surprisal and entropy,

**Fig. 1 | Illustration of theoretical framework, experimental paradigm, analysis, and expected results. A** Derivation of phoneme and acoustic edge features from the speech signal. **B** Experimental paradigm and analyzed datasets. **C** Outcomes of TRF analysis: model accuracy and model weights. **D** Expected outcomes about how acoustic edges and phoneme features are modulated by statistical familiarity and language structure.

derived from cohort model statistics (See "Methods"), reflect contextual modulation at the word level. Since we aimed to examine encoding differences between sensory acoustic information and abstract linguistic units, and phoneme-level statistical features also

capture information about categorized phonemes, we analyzed averaged responses to all phoneme features in the AC and STG. Instead of running separate whole-brain analyzes for each feature, this approach reduced computational demands when analyzing three

different datasets with nine different contrasts. This allowed us to investigate how linguistic structure modulates acoustic edges and phoneme features by comparing both model accuracy improvements and model weights, thereby examining the temporal dynamics of feature-specific neural contributions (See Fig. 1).

## Results

### Sentences vs word lists

At first, we compared the neural tracking of phoneme features (such as phoneme onset, surprisal, and entropy) and acoustic edges (acoustic onset spectrogram), which represents the sudden changes in the gammatone spectrogram of the acoustic signal, in both sentences and word lists using Dutch and Turkish stimulus datasets with native speakers.

To evaluate if acoustic edges, phoneme and word features significantly contributed to the accuracy, the accuracy differences of each subject between the full model which has all features and the model without the feature of interest were first averaged over AC and STG sources and then tested against zero with a $t$ test. Acoustic edges, phoneme and word features of both word list and sentence conditions significantly contributed to the accuracy, both in Dutch and Turkish-language datasets (Supplementary Tables 1 and 2).

We then fitted a linear mixed effect model (LME) with factors feature (acoustic edges and phoneme features) and condition (sentences and word lists) to compare the accuracy improvement in both languages separately. We used lme4 package to fit LMEs and reported the output of the lmerTest package's anova() function for the fitted model to obtain Type III F-statistics with Satterthwaite-approximated degrees of freedom and demonstrate the main and interaction effects.

For the Dutch language, we found that the model, which included the feature and condition interaction with a random slope for subjects, had a higher Bayesian information criterion (BIC) compared to the model without this interaction but the difference was not significant (BIC −869.12 vs −869.64, $\Delta\chi^2 = 1.48$, $p < 0.2244$).

There was a significant main effect of feature with stronger accuracy improvement for the acoustic edges compared to the phonemes ($F(1,55) = 94.67$, $p < 0.0001$). We used the "lsmeans" function in $R$[68] as a pairwise analysis to contrast the accuracy improvement by conditions for each feature and the accuracy improvement by features for each condition. This analysis showed that for both acoustic and phoneme features, there was not a significant accuracy improvement difference for the sentences compared to the word lists (acoustic edges: $t(55) = 0.296$, $p = 0.9909$; phonemes: $t(55) = 0.296$, $p = 0.9909$). For both sentences and word lists conditions, there was a stronger accuracy improvement by acoustic edges than the phoneme features (sentences: $t(55) = 9.730$, $p < 0.0001$; word list: $t(55) = 9.730$, $p < 0.0001$) (Fig. 2A). LME results are shown in Supplementary Table 3.

Then, for Turkish stimuli, we compared the models with and without an interaction effect, and the model without the interaction with random slope for subjects and feature had a lower BIC, but this was not significant (BIC −1615.5 vs −1616.0, $\Delta\chi^2 = 1.4917$, $p = 0.222$). There was a significant main effect of condition with a stronger accuracy improvement for sentences ($F(59) = 6.68$, $p = 0.0122$), but no effect of feature or interaction between them. Pairwise comparison showed that for both acoustic edges and phonemes, there was a marginally stronger accuracy improvement by sentence condition (acoustic edges: $t(59) = 2.585$, $p = 0.0576$; phonemes: $t(59) = 2.585$, $p = 0.0576$) (Fig. 2B). LME results are shown in Supplementary Table 4.

Next, we investigated the TRF weights to understand the timing at which each feature (acoustic edges and phoneme features) differentially contributed to the accuracies of encoding models under each condition. We compared the sum of absolute TRF weights at each source point between −50 ms and 700 ms relative to feature onset. A cluster-based permutation test was used to correct for multiple comparisons. In Dutch stimuli, the analysis of TRF weights of phoneme features in the sentences condition showed a stronger response than word list condition ($p < 0.05$) in all time points. In the Turkish dataset, phoneme weights compared to acoustic edge

weights were greater between 350 and 550 ms but smaller between −50 and 110 ms. However, no significant difference was found in TRF weights for acoustic edges in both datasets (Fig. 2C, D).

Although the improvement in prediction accuracy due to acoustic edges and phoneme features did not differ significantly between the sentence and word conditions in either dataset, the TRF weights exhibited distinct patterns across datasets. In the Dutch dataset, phoneme feature weights for sentences were greater than those for words throughout the entire time interval. In contrast, the Turkish dataset showed the opposite pattern during the early time window (−30 to 140 ms), but phoneme feature weights were again higher for sentences than for words in the later time window (370 to 570 ms). This difference in TRF weights may stem from variations in the contextual information of the stimuli or in the methods used to generate them. Each sentence had exactly 10 words in Dutch stimuli and 4 words in Turkish stimuli (see "Methods"). We compared the predictability of each word in sentences in Dutch and Turkish stimuli by calculating the surprisal values of each word using a GPT2 model (Dutch[69]; Turkish[70]). Words in Dutch sentences had a lower surprisal value (more predictable) than words in Turkish stimuli ($p < 0.0001$, mean surprisal Dutch words = 13.16, std = 5.46; mean surprisal Turkish words = 14.33, std = 3.79) possibly due to a longer context - more words in a sentence - in Dutch stimuli. In the Turkish dataset, the words were isochronous, meaning that the onset of each word was perfectly predictable. In contrast, the Dutch dataset contained naturally spoken words with variable onset times and durations. We discussed how these differences might have contributed to the divergent results in the "Discussion" section.

### Words vs random-syllable streams

We then compared the accuracy improvements of phoneme features and acoustic edges in words and random syllables streams in Dutch, Mandarin Chinese and Turkish stimuli. Mandarin Chinese and Dutch stimuli were listened to by both Dutch-speaking and Mandarin-speaking participants, who could not comprehend the non-native languages presented to them. Turkish stimuli were listened to by only Turkish-speaking participants because the data were collected in a different experiment. All stimuli for all languages were isochronous synthesized speech with equal syllable duration. Random syllables streams condition was generated by shuffling the order of syllables in words condition (see "Methods").

Both acoustic edges and phoneme features of Dutch, Chinese and Turkish stimuli significantly contributed to the accuracy in word lists and random syllable streams condition for all participant groups (Supplementary Tables 5–7).

We then fitted a LME to compare the model accuracy improvement by acoustic edges and phoneme features in words and random syllables streams conditions for all stimuli. We compared the models with and without an interaction effect and the model with the interaction has a significantly lower BIC for almost all cases tested except when Dutch speaking participants were listening to Mandarin Chinese stimuli (Mandarin-speaking participants—Mandarin Chinese stimuli: BIC −564.76 vs −578.84, $\Delta\chi^2 = 16.07$, $p < 0.0001$; Mandarin-speaking participants—Dutch stimuli: BIC −587.16 vs −619.41, $\Delta\chi^2 = 38.25$, $p < 0.0001$; Dutch-speaking participants—Mandarin Chinese stimuli: BIC −770.56 vs −763.85, $\Delta\chi^2 = 1.29$, $p = 0.863$; Dutch-speaking participants—Dutch stimuli: BIC −593.46 vs −602.05, $\Delta\chi^2 = 10.59$, $p = 0.0011$; Turkish-speaking participants—Turkish stimuli: BIC −1132.8 vs −1221.6, $\Delta\chi^2 = 94.73$, $p < 0.0001$).

Besides this interaction (Mandarin-speaking participants—Mandarin Chinese stimuli: $F(1,26) = 20.16$, $p = 0.0001$; Mandarin-speaking participants—Dutch stimuli: $F(1,26) = 39.75$, $p < 0.0001$; Dutch-speaking participants—Dutch stimuli: $F(1,42) = 11.14$, $p = 0.0001$; Turkish-speaking participants—Turkish stimuli: $F(1,87) = 98.42$, $p < 0.0001$), there was a significant main effect of feature with a stronger model accuracy improvement for the acoustic edges compared to the phonemes when Dutch-speaking participants listening to Mandarin Chinese ($F(1,14) = 24.87$, $p = 0.0001$), and with a stronger model accuracy improvement for the phonemes compared to the acoustic edges when Turkish stimuli presented

**Fig. 2 | Accuracy improvement and TRF weights.**
**A, B** Accuracy improvement averaged over all
sources on both right and left AC and STG for
sentences and word lists in Dutch and Turkish sti-
muli, respectively. Boxes indicate the interquartile
range (IQR), with the median shown as a black
horizontal line. Whiskers extend to 1.5×IQR.
**C, D** Average of the absolute values of weights over
all sources on both right and left AC and STG and
each time lag for sentences and word lists in Dutch
and Turkish stimuli, respectively. Shaded area
indicates the standard error of the mean. Red lines
under the graphs show the time points where there is
a significant difference between conditions.
($^{****}$<0.0001, $^{***}$<0.001, $^{**}$<0.01, $^{*}$< 0.05).

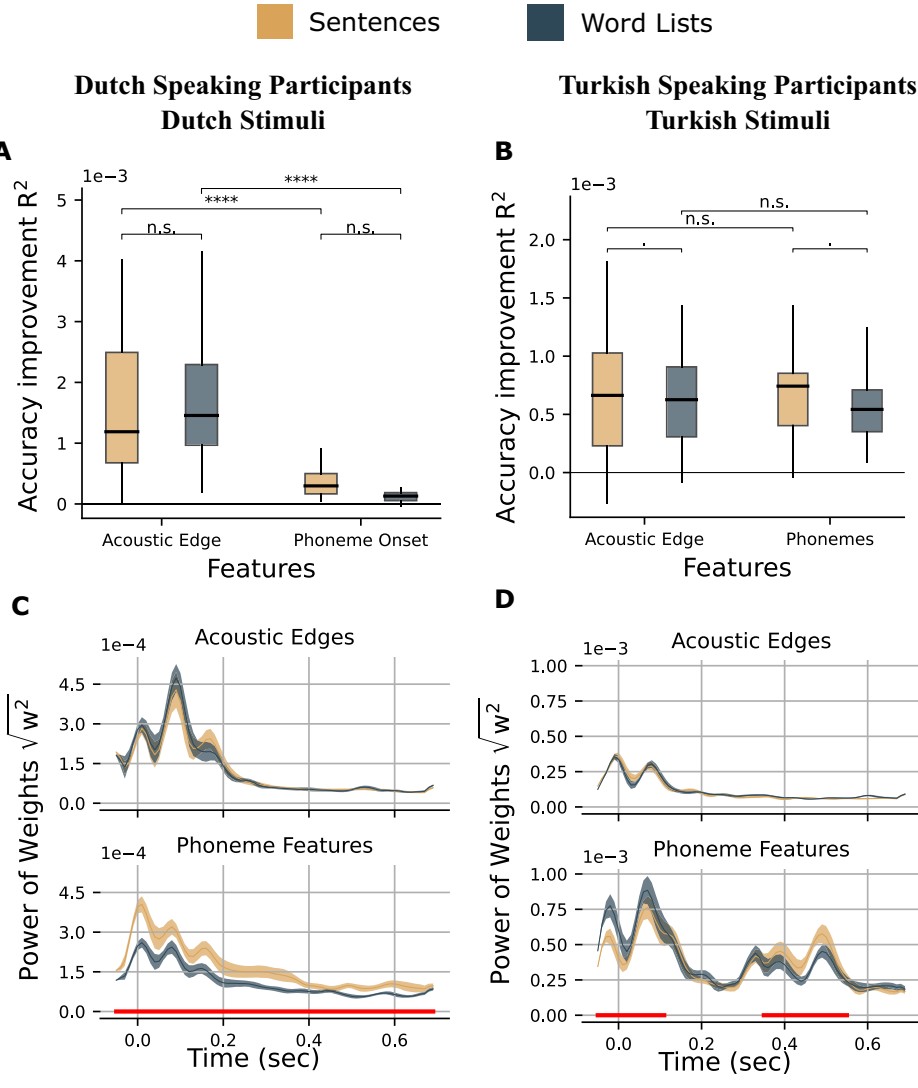

to Turkish-speaking participants ($F(1,30) = 67.23$, $p < 0.0001$), and a sig-nificant main effect of condition, with a greater accuracy for the word lists streams compare to the random syllable streams in all cases (Mandarin-speaking participants—Mandarin Chinese stimuli: $F(1,26) = 13.37$, $p = 0.0011$; Mandarin-speaking participants— Dutch stimuli: $F(1,26) = 44.61$, $p < 0.0001$; Dutch-speaking participants—Mandarin Chi-nese stimuli: $F(1,29) = 19.80$, $p = 0.0001$; Dutch-speaking participants—Dutch stimuli: $F(1,42) = 13.32$, $p = 0.0007$; Turkish-speaking participant—Turkish stimuli: F Value = 110.61, $p < 0.0001$).

Pairwise comparison showed that for the phoneme feature, there was a stronger accuracy improvement for the word lists streams compared to the random syllable streams (Mandarin-speaking participants—Mandarin Chinese stimuli: $t(26) = 5.76$, $p < 0.0001$; Mandarin-speaking participants—Dutch stimuli: $t(26) = 9.18$, $p < 0.0001$; Dutch-speaking participants—Mandarin Chinese stimuli: $t(28) = 3.84$, $p = 0.0034$; Dutch-speaking parti-cipants—Dutch stimuli: $t(42) = 4.94$, $p = 0.0001$; Turkish-speaking partici-pants—Turkish stimuli: $t(58) = 14.45$, $p < 0.0001$). This difference was not significant for the acoustic edges.

For word list streams, there was a stronger accuracy improvement by the phoneme features than the acoustic edges when Dutch and Turkish-speaking participants were listening to their native language, but there was no significant difference between them when Chinese-speaking participants were listening to native and nonnative language. On the contrary, accuracy improvement by the acoustic edges was stronger when Dutch-speaking

participants were listening to nonnative language. (Mandarin-speaking participants—Mandarin Chinese stimuli: $t(21.1) = 0.61$, $p = 0.9269$; Mandarin-speaking participants—Dutch stimuli: $t(21.7) = 1.93$, $p = 0.2441$; Dutch-speaking participants—Mandarin Chinese stimuli: t(19.2) = −4.17, $p = 0.0026$; Dutch-speaking participants—Dutch stimuli: $t(42) = 2.98$, $p = 0.0234$; Turkish-speaking participants—Turkish stimuli: $t(42.7) = 11.68$, $p < 0.0001$).

For random syllables streams, accuracy improvement by the phoneme features was smaller than the acoustic edges when participants were lis-tening to Dutch and Mandarin Chinese stimuli except when Dutch-speaking participants were listening to Dutch syllables (Mandarin-speaking participants—Mandarin Chinese stimuli: $t(21.1) = 3.73$, $p = 0.0062$; Mandarin-speaking participants—Dutch stimuli: $t(21.7) = 4.31$, $p = 0.0016$; Dutch-speaking participants—Mandarin Chinese stimuli: $t(19.2) = 4.96$, $p = 0.0005$; Dutch-speaking participants—Dutch stimuli: $t(42) = 1.73$, $p = 0.3187$). On the other hand, accuracy improvement by the phoneme features was stronger when listening to Turkish stimuli (Turkish-speaking participants—Turkish stimuli: $t(42.7) = −3.10$, $p = 0.0171$) (Fig. 3). LME results are shown in Supplementary Tables 8–12.

Next, we compared the TRF weights of features in words and random syllables streams with a cluster-based permutation test. In all stimuli, the analysis of TRF weights of phoneme features in the words streams condition showed a stronger response than the random syllables streams condition ($p < 0.05$). The difference was significant in all time points for most

participants except when Dutch-speaking participants were listening to Mandarin Chinese stimuli. For them, it was between 220 and 700 ms. However, no significant difference was found in TRF weights for acoustic edges in most of the cases, except for Turkish-speaking participants. TRF weight of acoustic edges in syllables condition was stronger between 30 and 170 ms (Fig. 4).

## Familiarity with the uncomprehended language

In the first section, Sentences and Word lists are presented to native speakers only, and the number of presentations of the same word was equal for both conditions. Enhancement of phoneme-feature tracking should thus be driven by linguistic structure. For the Words vs Syllable streams contrast in Dutch and Mandarin Chinese stimuli, we cannot conclude that any observed effects could be driven only by the word structure, because transitional probabilities between syllables are higher in the "words" condition compared to the "random syllables" condition, as the syllable sequences in the latter are fully randomized. In contrast, the "words" condition features consistent syllable pairings, where a specific syllable reliably follows another, preserving structured statistical regularities. So, we also compared the neural tracking of acoustic and phoneme features of words in a native vs a familiar uncomprehended (Dutch for Mandarin-speaking participants) and unfamiliar uncomprehended language (Mandarin Chinese for Dutch-speaking participants). It is important to note that this analysis pertains to the same dataset as previously examined but focuses on the aspect of language familiarity, which was not assessed in the earlier analysis. In the condition where participants are exposed to a familiar but incomprehensible language, they are already acquainted with the statistical regularities of the language. In contrast, in the unfamiliar and incomprehensible language condition, participants have the opportunity to learn these statistical patterns during the experiment. When comparing these conditions to their native language, we expect to observe the influence of word structure and lexical semantics derived from participants' pre-existing linguistic knowledge. This influence would extend beyond the statistical information acquired during the experiment and their everyday linguistic experience.

We fitted a LME model for words to evaluate the effect of language and nonnative language familiarity (as Dutch speaker participants were not familiar with Mandarin Chinese, but Mandarin Chinese speaker participants were familiar with Dutch) on the accuracy improvement of features for both Mandarin-speaking and Dutch-speaking participant groups. We compared the models with and without an interaction effect and for participant groups (Mandarin and Dutch Speakers), stimuli (Native and Nonnative) and feature (Acoustic Edges and Phoneme features), and the model with the interaction has a lower Bayesian information criterion (BIC $-1165.7$ vs $-1170.2$, $\Delta\chi^2 = 12.55$, $p = 0.0137$). We found a significant interaction between Group, Stimuli and Feature ($F(1,81) = 6.29$, $p = 0.0141$). Pairwise analysis showed that only for the Dutch participants group, accuracy improvement by phoneme features was stronger in native stimuli compared to nonnative stimuli ($t(81) = 4.01$, $p = 0.0031$; Fig. 5). LME results are shown in Supplementary Table 13.

Next, we compared the TRF weights of features in words streams between languages in both participant groups with a cluster-based permutation test. For Mandarin-speaking participants, we found a significant difference in the acoustic edge weights of Mandarin Chinese and Dutch stimuli between 0 and 250 ms and in phoneme features between 100 and 700 ms (Fig. 5C). For Dutch-speaking participants, phoneme features weights were significantly different between $-50$ and 700 ms, and the difference was significant for acoustic edges between $-50$ and 100 ms (Fig. 5D).

We also compared the latencies of Dutch and Chinese stimuli peaks on early ($-50$ and 100 ms) and mid (100–350 ms) time interval for acoustic edges and on late time interval (350–700) for phonemes (Brodbeck et al.[7]). Since the peak times were not normally distributed we conducted an ANOVA (Analysis of Variance) using aligned rank transformation (ART) with the artlm() function from the ARTool package in R. The model included the main effects of Stimuli (Native vs. Nonnative), Group (Dutch

vs. Mandarin speakers), Feature (Acoustic Edge vs. Phoneme), and Time (Early, Mid, Late).

This analysis revealed significant main effect of Stimuli, with earlier peak times for native languages ($F(1,324) = 15.23$, $p = 0.0001$), main effect of Feature, with earlier peak times for acoustic edges ($F(1,324) = 5.81$, $p = 0.0165$), and main effect of Group ($F(1,324) = 3.92$, $p = 0.0486$), with the Mandarin group showing earlier peak times. We also observed a significant three-way interaction among Stimuli, Feature, and Time ($F(2,324) = 3.46$, $p = 0.0327$) and among Stimuli, Group, and Time ($F(2,324) = 3.66$, $p = 0.0268$).

Pairwise comparisons of native versus non-native stimuli for each combination of Feature, Group, and Time showed only one marginally significant effect: for Mandarin speakers, phoneme features in the late interval exhibited earlier peak times for native language stimuli ($t(324) = -1.83$, $p = 0.068$).

## Discussion

In this study, we examined the influence of varying levels of linguistic structure (syllables, words, and sentences) in the absence of predictability regarding the subsequent word from the broader discourse context, because sentences (or words in word streams condition) are contextually independent of each other. Our findings indicate that phoneme feature weights within sentences were greater compared to word lists in a comprehended language, also an enhancement of phoneme feature weights and model accuracies was observed in words compared to random syllables in both comprehended and uncomprehended languages[5,39,71]. We observed a significant difference in the weights of acoustic edges only in Turkish stimuli with greater weights of acoustic edges in syllables compared to words. In contrast between sentences and word lists in comprehended languages, the enhancement of phoneme features is likely associated with the integration of words into a sentence structure[49,51,72]. In the contrast between words and syllable streams, the enhancement of phoneme features occurs independently of comprehension, possibly due to both statistical learning[62,63,73] and structure building. This enhancement is more pronounced when participants are already familiar with the uncomprehended language. When we examined the effects of language familiarity in an uncomprehended language, we found that phoneme feature encoding was enhanced to the same level as in the native language in Mandarin-speaking participants who were already familiar with nonnative language, but it was not the case for unfamiliar uncomprehended language. The weight of the model showed an earlier peak in the native language than uncomprehended language although the model accuracies were at the same level, which might be due to earlier recognition of words in the native language. Additionally, in the uncomprehended language, acoustic edge weights of word streams were stronger than in the native language in the early time interval. This could be related to processing of acoustic information, where acoustic edges still provide valuable information for processing uncomprehended words or statistical patterns of speech sounds. In contrast, in the native languages, acoustic information at the early stage is suppressed when words are recognized which is also observed in the words and random syllable comparison in the Turkish dataset and the previous studies[34,35,38,40]. Our findings point more broadly to an interplay between knowledge of the structure of language and statistical experience with speech contributing to the strength of neural encoding of information during speech and language processing (See Fig. 6).

In evaluating the influence of sentence structure on neural processing, we contrasted sentences with word lists by analyzing acoustic edge and phoneme feature tracking across two distinct datasets. Both datasets demonstrated an enhancement in the TRF weights for phoneme features in the sentence condition, which indicates an increase in the amplitude of the predicted signal by the model. However, this enhancement was not significant in model accuracies, which measures the alignment between the phases of the neural and predicted signals. Incorporating word-level features significantly improved model accuracy, yet the contrast between the

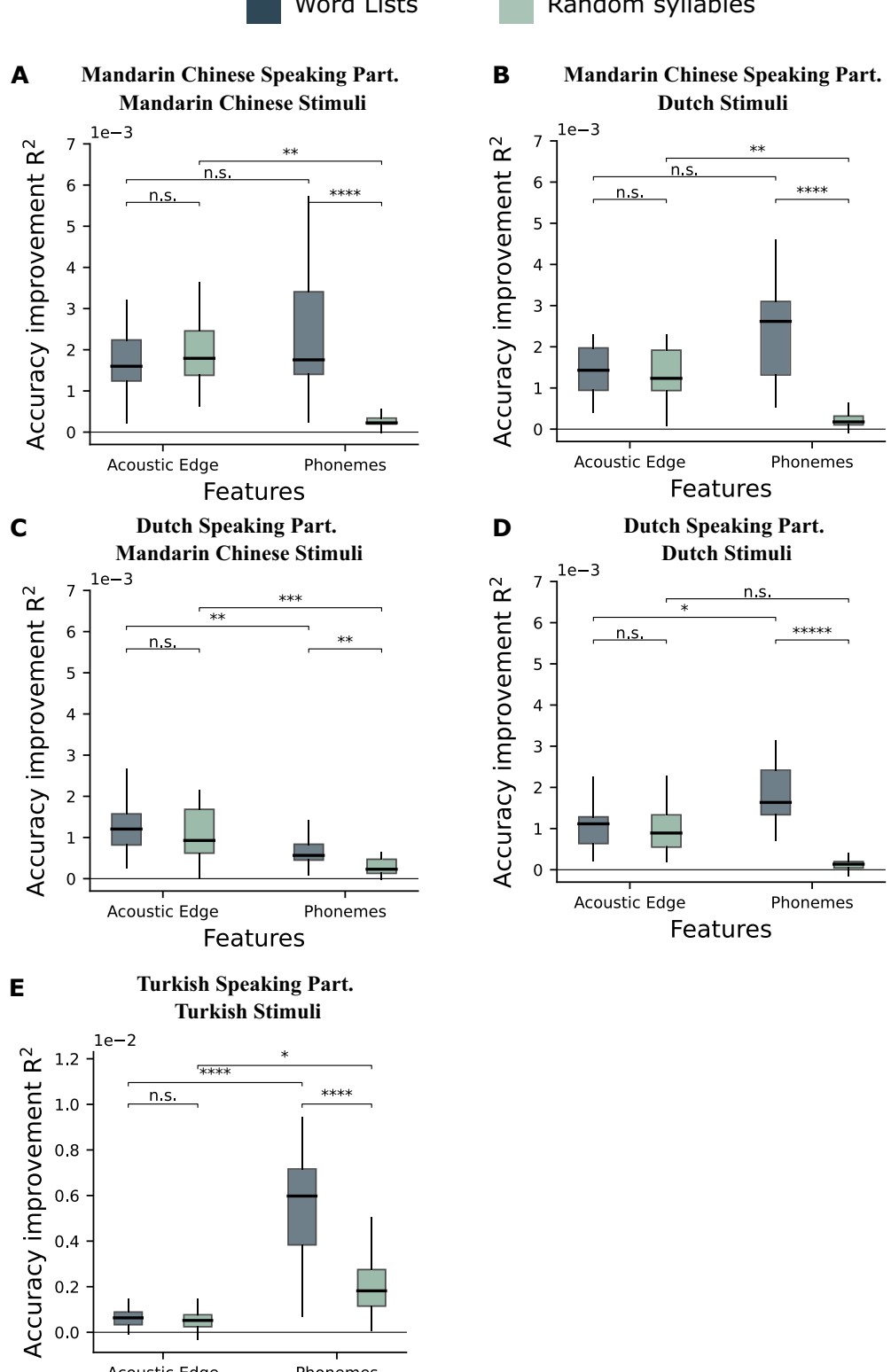

**Fig. 3 | Accuracy improvement. A** Mandarin-speaking participants listening to Mandarin Chinese stimuli, **B** Mandarin-speaking participants listening to Dutch stimuli, **C** Dutch-speaking participants listening to Mandarin Chinese stimuli, **D** Dutch-speaking participants listening to Dutch stimuli, **E** Turkish-speaking participants listening to Turkish stimuli. Accuracy improvement averaged over all sources on both right and left AC and STG for words and syllables. Boxes indicate the interquartile range (IQR), with the median shown as a black horizontal line. Whiskers extend to 1.5×IQR. ($^{****}$<0.0001, $^{***}$<0.001, $^{**}$<0.01, $^{*}$< 0.05).

**Fig. 4 | TRF weights. A** Mandarin-speaking participants listening to Mandarin Chinese stimuli, **B** Mandarin-speaking participants listening to Dutch stimuli, **C** Dutch-speaking participants listening to Chinese stimuli, **D** Dutch-speaking participants listening to Dutch stimuli, **E** Turkish-speaking participants listening to Turkish stimuli. Average of the absolute values of weights over all sources on both right and left AC and STG and each time lag for words and random syllables. Shaded area indicates the standard error of the mean. Red lines under the graphs show the time points where there is a significant difference between conditions. (****<0.0001, ***<0.001, **<0.01, *< 0.05).

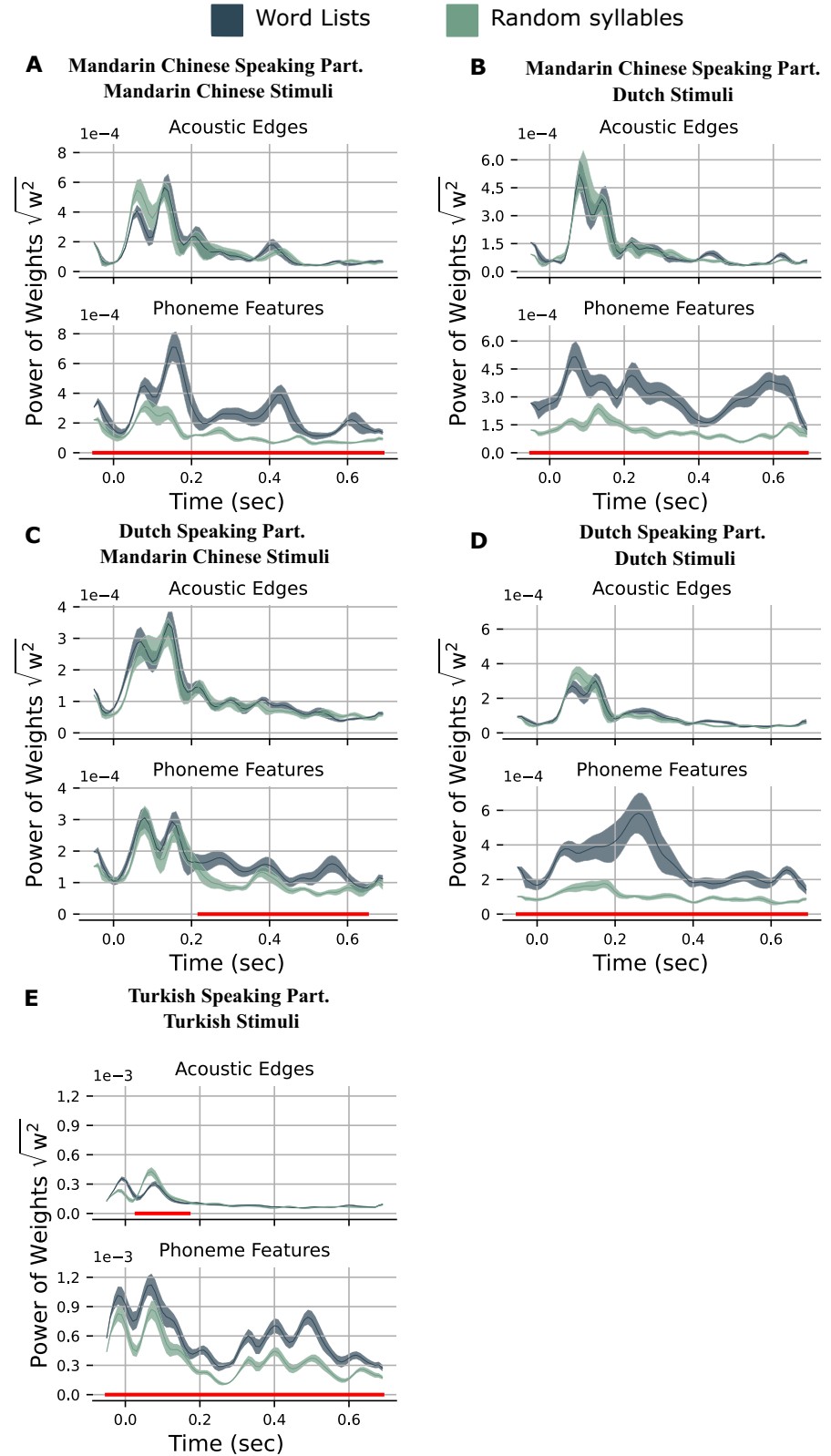

The encoding of acoustic edges exhibited no significant difference between sentences and word lists. This finding aligns with the results reported by Gillis et al. [5], who similarly found no difference in acoustic feature encoding between sentences and word lists. However, our previous study [38] demonstrated that both acoustic and phoneme features are attenuated when the predictability of the subsequent word increases. These results suggest a distinct modulatory effect of linguistic structure and predictability of next word so this discrepancy may stem from the dynamic modulation of acoustic and phoneme features by an integrative and a predictive process [74,75]. Specifically, within a sentence, the strength of acoustic

sentence and word-list conditions still demonstrated a higher TRF weights for phonemes in the sentence condition.

**Fig. 5 | Accuracy improvement and TRF weights.** **A**, **B** Accuracy improvement for Mandarin-speaking and Dutch-speaking participants averaged over all sources on both right and left AC and STG for words. Boxes indicate the interquartile range (IQR), with the median shown as a black horizontal line. Whiskers extend to 1.5×IQR. **C**, **D** Average of the absolute values of weights over all sources on both right and left AC and STG and each time lag for words. Shaded area indicates the standard error of the mean. Red lines under the graphs show the time points where there is a significant difference between languages. ($^{****}$<0.0001, $^{***}$<0.001, $^{**}$<0.01, $^{*}$< 0.05).

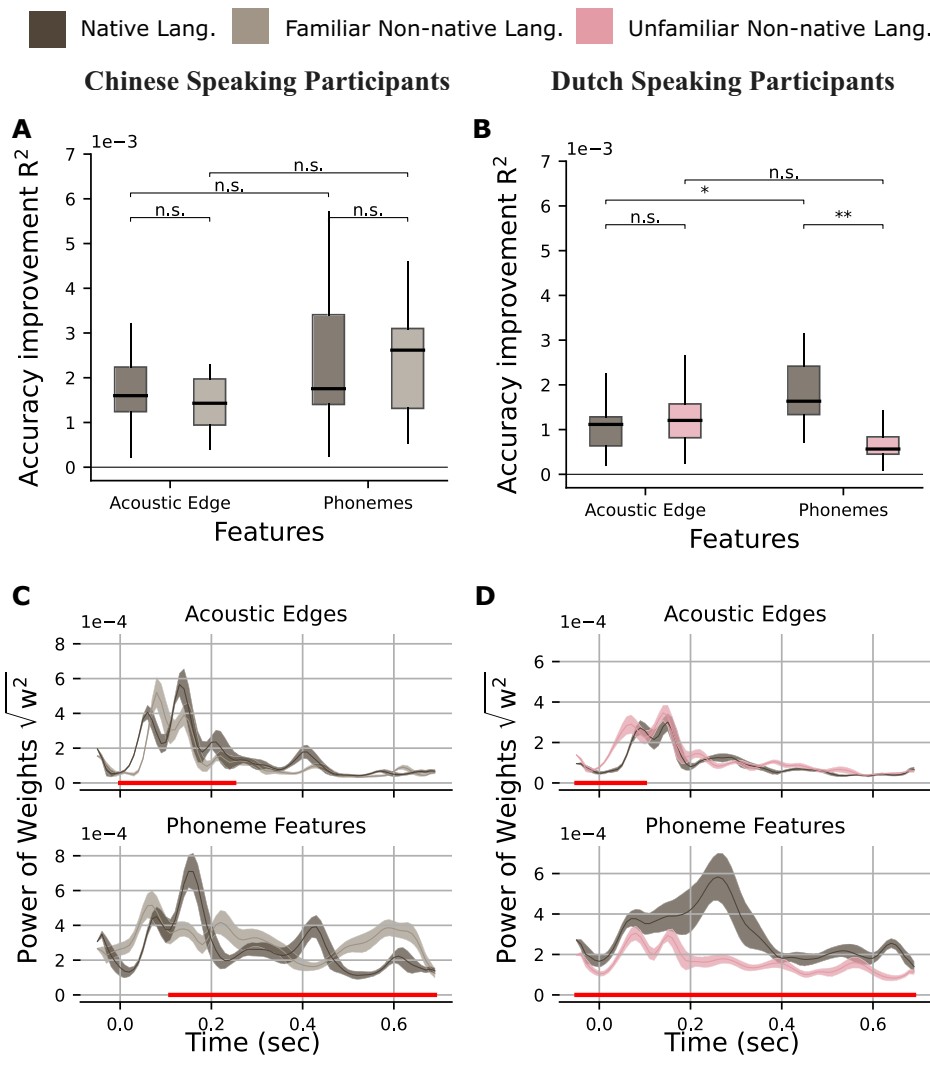

and phoneme feature encoding in each word may vary according to the preceding context, as indicated by the outcomes of our prior investigation. Phoneme features and acoustic edges may be suppressed when the next word is more predictable, or they may be enhanced when it is less predictable. On the other hand, when comparing the average encoding across all words in sentences and word lists, we observe a difference solely in phoneme feature encoding, which is more pronounced in sentences driven mainly by the integration processes of words into a sentence structure as the predictive process due to discourse context is less prominent compared to connected sentences in a story.

This suggests that phoneme feature enhancement is likely attributable to the integration of words into a cohesive sentence structure, guided by syntactic and semantic information[49,51,72]. This interpretation of the pattern of results is supported by improvement in the modeling of the neural response based on the inclusion of word-level statistical features such as word surprisal and entropy that reflect, to some degree, distributional information of syntactic and semantic content.

In addition, we observed a reduction in phoneme-feature weights during the early time interval in the Turkish dataset, an effect not present in the Dutch dataset. A key distinction between these two datasets is that, in the Turkish dataset, the stimuli consist of isochronous words. In the sentence condition, these words form repeating sentence structures, whereas in the words condition, they do not. This pattern may parallel the early-interval suppression of acoustic-edge responses previously reported for words compared to syllables, which has been attributed to perfectly predictable

timing and/or stimulus repetition. By analogy, the repetition of sentence structure or the highly predictable temporal patterning in the Turkish stimuli may suppress phoneme-related responses at early latencies. However, this interpretation remains speculative, and future studies are needed to directly test whether temporal regularity or structural repetition modulates phoneme-level encoding in this way.

We conducted a comparative analysis of acoustic and phoneme feature encoding between words and random syllable streams in both comprehended and uncomprehended languages to assess the influence of word structure. Across all languages examined (Dutch, Mandarin Chinese, and Turkish), irrespective of language comprehension, phoneme features in words streams were consistently encoded more strongly than in random syllables streams, while no significant difference was observed in acoustic edge model accuracies. These findings suggest that participants acquired knowledge about the statistical regularities of speech sounds in all languages during the experiment, particularly as words are repeatedly presented, leading to enhanced phoneme feature encoding, even in the absence of word- and language comprehension. For Turkish and Dutch-speaking participants when they are listening to words in their native languages, the encoding of phoneme features in words surpassed that of acoustic edges, suggesting an additional effect of language comprehension driven by integrating phonemes into word structure. On the other hand, when Dutch-speaking participants were listening to words in an unfamiliar nonnative language, acoustic edge tracking was stronger than phoneme feature tracking. There was not any significant difference between acoustic and

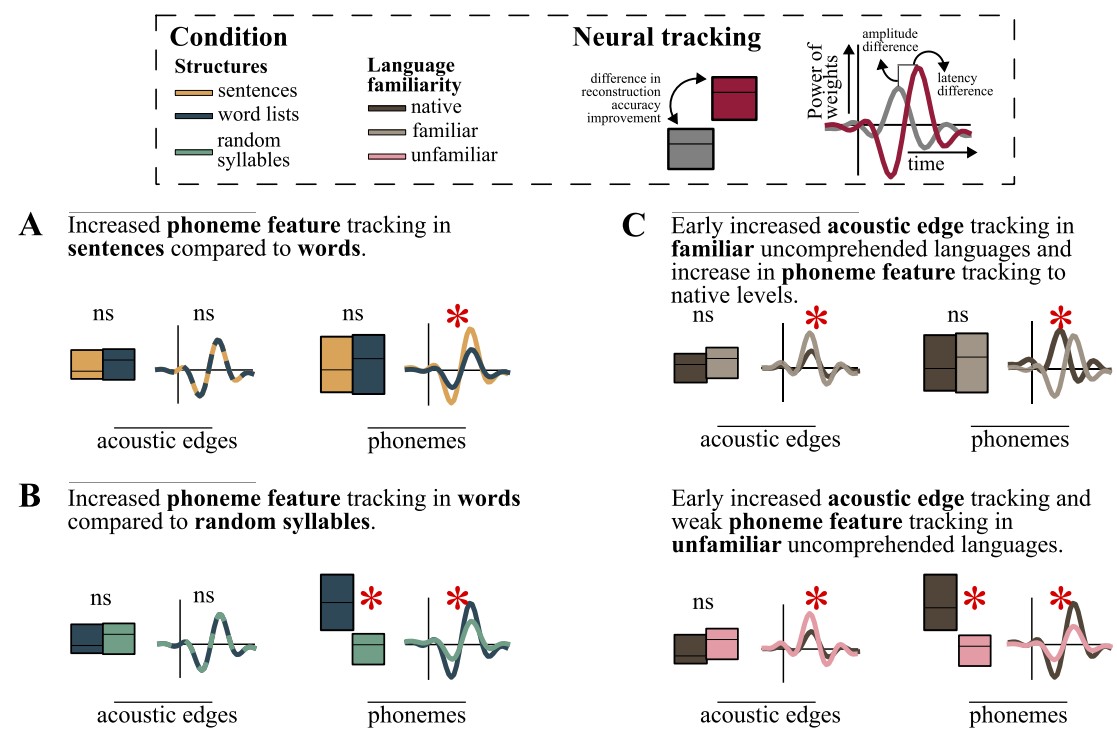

**Fig. 6 | Schematic summary of results. A, B** Linguistic structure modulates phoneme tracking but not acoustic edge tracking. **A** Sentences > words. Phoneme feature tracking is stronger in sentences than in word lists, as reflected in increased TRF weights. **B** Words > random syllables. Phoneme feature tracking is enhanced in word lists compared to random syllables, with increases in both TRF weights and model accuracy. **C** Language familiarity affects phoneme and acoustic edge tracking differently. In uncomprehended languages, TRF weights of acoustic edges increase. Phoneme tracking increases to native-like levels in familiar uncomprehended language with a delayed TRF weight peak. In unfamiliar language, phoneme tracking remains weak, as indicated by low TRF amplitudes and reduced model accuracy.

phoneme feature encoding for Mandarin-speaking participants for both native and nonnative languages. In naturalistic stimuli (Dataset 1 - Dutch sentences and in our previous study[38]), accuracy improvement by acoustic edges was stronger than by phoneme features. However, in the current study with repeated isochronous word streams, accuracy improvement by phonemes in the native language listening conditions was either stronger than acoustic edges (for Dutch and Turkish-speaking participants) or showed no significant difference (Mandarin-speaking participants). This could be related to the suppression of acoustic edges with repetition of isochronous stimuli in which the timing of the stimuli is perfectly predictable[76]. Acoustic edge suppression may be less noticeable in Mandarin-speaking participants, potentially because their native language is a tonal language[77], leading them to rely more heavily on acoustic cues.

The encoding of phoneme features showed a more striking contrast between words and random syllable streams in Dutch and Mandarin Chinese stimuli than in Turkish stimuli. This variation stems primarily from how random syllables were created across these datasets. In the Dutch and Mandarin Chinese sets, syllables were randomly shuffled, while in the Turkish set, random syllables were formed by combining the first syllable of one word with the second syllable of another. As a result, in the Turkish dataset, phoneme-level statistics of real words are disrupted only after the second syllable. Additionally, the Turkish dataset has less repetition of words and syllables compared to its Dutch and Mandarin Chinese counterparts, which could further explain the differences observed in acoustic edge weights between words and syllables in Turkish. Although a slight increase in acoustic edge weights appears in the syllables condition during an early time interval across all three native language listening conditions—Dutch, Mandarin Chinese, and Turkish—this difference is only statistically significant in the Turkish dataset.

The differential patterns we observe where word-syllable contrasts affect both accuracy and weights, while sentence-word contrasts affect only

weights likely reflect distinct underlying neural mechanisms. When processing random syllables versus words, the brain might be recruiting more distinct cortical circuits with different oscillatory signatures[78], resulting in differences in both phase alignment and power modulation (reflected in both accuracy and weights). In contrast, words presented in the word list condition versus within sentences might be utilizing the same network, but with altered temporal dynamics. The sentence structure might modulate how this network processes word-level information, adjusting the timing and strength of responses rather than recruiting different circuits (reflected only in weights).

Our previous analysis showed that both statistical exposure and language comprehension has an effect on increased phoneme feature tracking in words. To disentangle the effects of exposure and comprehension, we reanalyzed the data to contrast language familiarity and comprehension. We compared the encoding of acoustic edges and phoneme features among Dutch-speaking and Mandarin-speaking participants residing in the Netherlands when they were listening to word streams in both Dutch and Mandarin Chinese. Mandarin-speaking participants were familiar with Dutch, but Dutch-speaking participants were not familiar with Mandarin Chinese.

For Mandarin-speaking participants, no significant difference was observed in the phoneme feature encoding between Dutch and Mandarin Chinese stimuli. This finding suggests that when the uncomprehended language is familiar, the model accuracy of phoneme features during a statistical learning experiment can reach levels comparable to those of the native language. Mandarin-speaking participants were exposed to the Dutch language in daily life but could not speak or comprehend Dutch. Hence, this group of participants is likely to have acquired the statistical regularities of the speech sounds of Dutch language from daily exposure. Their familiarity with the uncomprehended language can elucidate the improvement in model accuracy by phoneme features in Dutch stimuli to the same extent as their native language.

Although the improvement in model accuracy by phoneme features for both languages was statistically indistinguishable for Mandarin-speaking participants, comparison of TRF weights of phoneme features revealed a stronger response around 437 ms for Mandarin Chinese stimuli and a later response (around 543 ms) for Dutch stimuli. In the native language, participants may have recognized words earlier due to a native mental lexicon, whereas for the non-native language, despite learning the statistical regularities of phonemes in the presented words during the experiment or daily life, listeners may need to continue to pay attention until the end of the word[49,79,80]. Another possible explanation for the delayed phoneme peak in the nonnative language is competition between newly lexicalized nonnative word forms (without semantic mappings) and existing native words. Prior research shows that repeated exposure to nonwords (in this case, words in an uncomprehended language) can lead to their lexicalization without meaning[81,82]. Because these newly formed lexical entries have weaker memory traces, they require more time to process[52,83,84]. It remains inconclusive whether this difference in weights between native and uncomprehended languages is purely driven by having a more accurate cohort model of the language or also involves mapping the word form to lexical meaning in the native language, as the experimental design of the analyzed dataset does not permit such a comparison. This poses an intriguing research question for future studies.

For Dutch-speaking participants who were unfamiliar with Mandarin Chinese, model accuracy of phoneme features in their native language is stronger than in the uncomprehended language. The model weights of phoneme features for native languages exceeded those for uncomprehended languages across the entire time interval. For Dutch-speaking participants, the strength of phoneme feature weights around 400 ms for the native language was not as pronounced as in the Mandarin-speaking participant group. This could be attributed to the fact that stimuli with isochronous syllables may be less natural for Dutch-speaking participants than for Mandarin-speaking participants, as Mandarin Chinese is a syllable-timed language[85,86] where syllables have similar durations, whereas Dutch is a stress-timed language[87,88]. To mitigate this effect, future studies may contrast familiar and unfamiliar uncomprehended languages by using fully naturalistic stimuli.

There was no significant difference in the improvement of model accuracy by acoustic edges between languages, but we observed a difference in acoustic edge weights between Dutch and Mandarin Chinese stimuli for both Dutch and Mandarin-speaking participants with smaller weights for their native language in the early time interval (between −50 and 100 ms), suggesting that participants might suppress acoustic edges for the comprehended language. This result is consistent with earlier studies that demonstrated the suppression of acoustic edges with increasing language proficiency[34,35,38,40]. A similar difference is also observed between words and syllables in the Turkish dataset, and a nonsignificant trend for Dutch and Mandarin speaking participants when they were listening to their native languages. For nonnative languages and when syllables didn't form words, acoustic edges may thus have been less suppressed as both language proficiency and abstract linguistic representations were lacking, and low-level acoustic information might be still informative.

## Conclusion

In this study, we examined how the presence of varying levels of linguistic structure (syllables, words, sentences) and the statistical experience with a spoken language influenced the encoding of acoustic and phoneme features both during and in the absence of language comprehension. Despite analyzing distinct datasets with varying presentation structures across three different languages, we consistently observed that phoneme features are more robustly encoded within sentences compared to isolated word lists in comprehended languages, and within words compared to random syllables, regardless of comprehension. This enhancement may be attributable to the integration of linguistic units into increasingly

abstract structures and points to the role of statistical information in the strength of neural representations. Our analyzes suggest that statistical information can be obtained either from the mental lexicon, in the case of native language comprehension, or from experience with a spoken language, in the absence of language comprehension. Even short-term repeated exposure to an uncomprehended spoken language, as in the experimental setups for the Dutch and Mandarin Chinese languages, can increase the tracking of phoneme features by the brain; yet the time dynamics of phoneme feature-tracking show distinct profiles in a native versus nonnative language. Acoustic edge tracking may be suppressed when lexical information is utilized during the recognition of words.

## Materials and methods
### Participants
*Dataset 1: Sentences vs Words in Dutch:* We analyzed the MEG dataset collected by Ten Oever et al.[37]. 20 Dutch native speakers (16 females; age range: 18–59; mean age = 39.5) participated in the study. One participant was excluded from the analysis as they did not finish the full session. *Dataset 2: Words and Random Syllables in Chinese and Dutch:* We analyzed the MEG Dataset collected by Bai, Meyer, te Rietmolen, and Martin (2022)[89]. Fifteen Dutch native speakers (9 females and 6 males), aged 20 to 35, and fourteen Mandarin Chinese native speakers (12 females; aged range 20–35) participated in the study. No participant was excluded from the analysis. For this study, we asked Mandarin Chinese-speaking participants how long were they living in the Netherlands at the time of the experiment, and we received a reply from 7 of them. On average, they had resided in the Netherlands for 24 months (s.d. 17.6 months). *Dataset 3: Turkish Sentences, Words and Random Syllables:* The study included 30 participants who were native Turkish speakers, consisting of 17 females. The participants' ages ranged from 22 to 42, with a mean age of 32.4. One participant was excluded from the analysis due to the pseudo-randomized presentation order mistake of experimental conditions. All participants were recruited via posters placed in public areas (e.g., university) and suitable websites (e.g., university websites). Notably, the University of Radboud holds a website and keeps a participant database especially for recruiting participants. All participants were right-handed, reported normal hearing, had either normal vision or vision corrected to normal, and had no history of dyslexia or other language-related disorders. Prior to participating in the MEG and MRI sessions, participants underwent a screening for eligibility and provided written informed consent. Participants were given monetary reimbursement for their participation. The study received approval from the Ethical Commission for Human Research in Arnhem/Nijmegen (project number CMO2014/288). Participants were compensated for their involvement in the study.

### Stimuli
*Dataset 1 - Sentences and Words in Dutch:* Stimuli consisted of naturally spoken sentences and word lists which consisted of 10 words (see Table 1 for examples). The sentences contained two coordinate clauses with the following structure: [Adj N V N Conj Det Adj N V N]. All words were bisyllabic except for the words 'de' (the) and 'en' (and). Word lists were word-scrambled versions of the original sentences which always followed the structure [V V Adj Adj Det Conj N N N N] or [N N N N Det Conj V V Adj Adj] to ensure that they were grammatically incorrect. In total, 60 sentences were used. All sentences were presented at a comfortable sound level (see[36,37] and Table 1 for examples).

*Dataset 2 - Words and Random Syllables in Chinese and Dutch:* To create the Dutch language materials, 20 bi-syllabic singular nouns were synthesized by the ReadSpeaker synthesizer (https://www.readspeaker.com/, the male voice, Guus), and then 40 syllables were extracted manually. Using the same method, 20 bi-syllabic nouns from Mandarin Chinese (which has no singular vs. plural distinction) were synthesized by Read-Speaker (the male voice, Liang), following which 40 syllables were extracted

to create the Mandarin Chinese language materials. In both languages, syllables were 153 to 302 ms (mean 230 ms) in duration. Each syllable was first resampled to 44.1 kHz, then adjusted to 250 ms by truncation or zero padding evenly at both ends. Five percent of both ends of each syllable was ramped by a cosine wave (see[89] and Table 2 for examples).

*Dataset 3 - Sentences, Words and Random Syllables in Turkish:* Tezcan et al.[90] generated 320 bi-syllabic Turkish words, comprising 80 adjectives, subjects, objects, adverbs, and verbs each, intended for use in the condition trials using the Google Cloud Text-to-Speech service (Google Cloud, 2023 with Female voice (tr-TR-wavenet-A). Additionally, 160 bi-syllabic buffer words were generated for use in the task trials and before-after condition trials. The sound waves were divided into two syllables using Praat Software[91] manually. Subsequently, the duration of each syllable was adjusted to 200 ms through stretching or compressing the sound wave using the Time-Scale Modification algorithm, which alters the length of an audio signal while preserving its pitch[92]. Additionally, the sound level was normalized to −15 dBFS. The Control Sentences condition comprised 16 sentences (64 words) in each trial, presented consecutively without any silence between them. Each sentence consisted of 4 words in the order of Adj-Subj-Noun/Adv-Verb, forming a meaningful and grammatical sentence. The Random Word List condition was created by shuffling the order of words in each sentence from the Control Sentences condition to prevent the formation of a grammatical sentence. For the Random Syllables condition, pseudowords were generated by shuffling the order of the first and second syllables of real words. Pseudowords consistently began with the first word of a real word and concluded with the second syllable of another real word. Similar to the other conditions, there were 20 sequences, each containing 64 pseudowords (see[90] and Table 3 for examples).

## Data acquisition

Brain activity of participants was recorded using magnetoencephalography (MEG) with a 275-sensor axial gradiometer system (CTF Systems Inc.) in a magnetically shielded room. All stimuli were presented audibly by using the Psychophysics Toolbox extensions of Matlab[93–95] while participants were fixating a cross in the middle of the presentation screen. MEG data were acquired at a sampling frequency of 1200 Hz. Head localization was monitored during the experiment using marker coils placed at the cardinal points of the head (nasion, left and right ear canal) and head position was corrected before each story part presentation to keep it at the same position as at the beginning of the experiment. In addition to MEG data, we also acquired T1-weighted structural MR images using a 3T MAGNETOM Skyra scanner (Siemens Healthcare, Erlangen, Germany). Lastly, 3-dimensional coordinates of each participant's head surface were measured using a digitizing pen system (Polhemus Isotrak system, Kaiser Aerospace Inc.)

## Procedure

*Dataset 1 - Sentences and Words in Dutch:* Ten Oever et al.[37] asked participants to perform four different tasks on these stimuli: a passive task, a syllable task, a word task, and a word-combination task. We only analyzed the MEG data during the word task to be consistent with the Turkish dataset. For the word task, two words were displayed on the screen after each trial (a random word from the just presented sentence and one random word from all other sentences excluding 'de' and 'en'), and participants needed to indicate which of the two words was part of the sentence before. Audio recordings were presented after a random interval between 1.5 and 3 s after the response of the participants; 1 s after the end of the audio, the task was presented. The block order was pseudo-randomized by separately randomizing both the task and the condition, so that they were counter-balanced among participants.

*Dataset 2 - Words and Random Syllables in Chinese and Dutch:* Bai, Meyer, te Rietmolen, and Martin (2022)[89] conducted six experiments with each participants group including two training experiments using a syllable recognition paradigm. The analysis in this study focuses specifically on the MEG data of Experiments 1 and 2, where stimuli were presented to participants before any training took place. To generate the stimuli for each participant, they randomly selected 10 singular nouns (20 syllables) from a pool of 20 words. Subsequently, a set of 100 noun sequences (word list condition) was stochastically concatenated, with each sequence lasting 4 s and containing eight singular nouns or 16 syllables. Following this, a set of 80 random syllable sequences was created by shuffling all the selected syllables. From this pool, 40 sequences were randomly chosen (random syllables condition, each with 16 syllables). During each trial, participants listened to an isochronous random syllables or word list sequence. After a brief 2 or 3 s of silence, a syllable target was presented, and participants were required to indicate, by pressing a button, whether or not the target syllable had appeared in the preceding sequence. The subsequent trial was initiated between 2000 and 2800 ms (random jitter) after the participants provided their response. All of these sequences were pseudo-randomly arranged in six blocks with 30 sequences in each block, so that they were counterbalanced among participants. To ensure an equal length of data for each condition, only the first 40 sequences of the word list condition were analyzed.

*Dataset 3 - Sentences, Words and Random Syllables in Turkish:* In the experiment conducted by Tezcan et al.[90], there were five distinct sentence sequence trials in each condition, each repeated four times, resulting in a total of 20 trials for each condition presented in 10 blocks. Presentation order of experimental conditions was pseudo-randomized and counterbalanced among participants. Four buffer words, not included in the analysis, were presented both before and after each trial. Audio recordings were played after a random interval ranging from 2 to 3 s. To maintain participant engagement throughout the experiment, 2–4 questions were included in each block. These questions were randomly presented at the end of trials, requiring participants to choose the last word they heard from four options. To prevent disruption of sequences in conditions with the task, new sets of buffer word sequences, ranging from 10 to 44 words in length, were generated specifically for use during these tasks. The block order was pseudo-randomized by separately randomizing both the task and the condition, so that they were counterbalanced among participants.

## Table 1 | Stimuli sentences vs word lists in Dutch

| Sentence | [bange helden] [plukken bloemen] en de [bruine vogels] [halen takken] |
|---|---|
|  | *[timid heroes] [pluck flowers] and the [brown birds] [gather branches]* |
| Word list | [helden bloemen] [vogels takken] de en [plukken halen] [bange bruine] |
|  | *[heroes flowers] [birds branches] the and [pluck gather] [timid brown]* |

## Table 2 | Stimuli random syllables vs words in Chinese and Dutch

| Chinese | Words | [怀表][键盘][相机][电视][熨斗][衣柜][冰箱][吉他]… |
|---|---|---|
|  |  | [pocket watch][keyboard][camera][televison][iron][wardrobe][refrigerator][guitar]… |
|  | Syllables | [带][他][机][视][电][发][汽][斗][篮][盘][怀][熨][沙][相][球][吉] |
| Dutch | Words | [lawaai][tafel][seizoen][gordijn][weide][limoen][suiker][tarwe]… |
|  |  | [noise][table][season][curtain][meadow][lime][sugar][wheat]… |
|  | Syllables | [be][li][tar][ker][wei][fel][la][we][sui][moen][gor][mer][ha][waai][mer][tar]… |

**Table 3 | Turkish sentences, word lists and random syllables**

| Sentences | [sersem maymun suyu dökmüş][yanlı basın haber yaptı][mutlu adam radyo açar]… |
|---|---|
| | *[clumsy monkey water spilled][biased press news reported][happy man radio turned-on]*… |
| Word lists | [maymun sersem dökmüş suyu][yanlı yaptı haber basın][adam mutlu açar radyo]… |
| | *[monkey clumpsy spilled water][biased reported news media][man happy turned-on radio]*… |
| Random syllables | [güner mayran güta alek][şisın raddi mutşil yedan][solgın kamun damniş parçer]… |

## MEG data preprocessing

MEG data were analyzed using mne-python (version 0.23.1). First, data were annotated to exclude the response parts from the rest of the analysis, then filtered between 0.5 and 100 Hz with one-pass, zero-phase, non-causal FIR filter using the default settings of mne-python. The data were resampled to 300 Hz and ocular and cardiac artifacts were removed with independent components analysis. Each trial was cropped −50 ms before the onset and 700 ms after the offset, then low-pass filtered at 8 Hz with one-pass, zero-phase, non-causal FIR filter, and then source localized.

Individual head models were created for each participant with their structural MR images with Freesurfer (surfer.nmr.mgh.harvard.edu) and were co-registered to the MEG coordinate system with mne coregistration utility. A surface-based source space was computed for each participant using fourfold icosahedral subdivision. Resting state data before the presentation of each trial were used to calculate the noise covariance matrix. Cortical sources of the MEG signals were estimated using noise-normalized minimum norm estimate method, called dynamic statistical parametric map. Orientations of the dipoles were constrained to be perpendicular to the cortical surface. Resulting source space is morphed to fsaverage template provided by FreeSurfer and the sources on auditory cortex (AC: Heschl's gyrus - G_temp_sup-G_T_transv) and superior temporal gyrus (STG: G_temp_sup-Lateral, G_temp_sup-Plan_polar, G_temp_sup-Plan_tempo) were extracted by using aparc.a2009s atlas implemented in mne-python[96]. Lastly, source time courses were resampled at 100 Hz.

## Predictor variables

### Acoustic features

Acoustic features were generated using the Eelbrain toolbox[97]. This included an 8-band gammatone spectrogram and an 8-band acoustic onset spectrogram, which models sudden changes in the gammatone spectrogram of the audio signal using an acoustic edge detection model[98]. Both spectrograms covered frequencies from 20 to 5000 Hz in equivalent rectangular bandwidth space.

### Phoneme onsets

For Dutch stimuli, phoneme onsets were extracted from the audio files of the trials automatically using the forced alignment tool from WebMAUS Basic module of the BAS Web Services[99,100]. For Mandarin Chinese and Turkish stimuli phoneme onsets were manually added by using Praat Software[91].

### Phoneme surprisal and entropy

Probabilities of each phoneme in a given word were calculated according to the probability distribution over the lexicon of each language weighted by the occurrence frequency of each word. When each phoneme unfolds in a word, it reduces the number of possible words in the cohort and generates a subset of cohort$_i$. Conditional probability of each phoneme Ph$_i$ given the previous phoneme equals to the ratio of total frequencies of the words in the remaining cohort to previous cohort.

$$P\left(Ph_{(i-1)}\right) = \frac{\sum_{word \in cohort_i} freq_{word}(i)}{\sum_{word \in cohort_{(i-1)}} freq_{word}(i-1)}$$

The surprisal of phoneme Ph$_i$ is inversely related to the likelihood of that phoneme.

$$S\left(Cohort_{(i)}\right) = -log_2(P(Ph_i))$$

The entropy of phoneme Ph$_i$ quantifies the uncertainty about the next phoneme Ph$_{i+1}$. It is calculated by taking the average of expected surprisal values of all possible phonemes.

$$E\left(Cohort_{(i)}\right) = \sum_{Ph}^{All\ phonemes} -P\left(cohort_{(i-1)}\right) * log_2\left(P\left(cohort_{(i-1)}\right)\right)$$

To calculate the probabilities of phonemes, we used SUBTLEX-NL dictionary[101] for Dutch stimuli, SUBTLEX-CH dictionary[102] for Chinese stimuli, and for Turkish stimuli Leipzig Corpora Collection Turkish dictionary[103] extracted from Wikipedia dumps. All dictionaries were filtered to eliminate the words which contains nonalphabetic characters.

## Sample size calculation

Sample size estimation was performed using the TTestPower function from the *statsmodels* library (version 0.12.2). Effect size was derived from a prior study reporting a model accuracy improvement based on phoneme features[38], calculated as the ratio of the observed mean difference to the standard deviation. A statistical power of 90% and a significance level ($\alpha$) of 0.05 were used for the calculation. The results indicated that a sample size of approximately 8 participants would be sufficient to detect statistically significant effects under these parameters.

## Linear encoding models

Contribution of each feature to the model accuracy is calculated by subtracting the model accuracy of a model that does not have that specific feature from the full model which has all features. Details of the model names and features included in models are shown in Supplementary Tables 15 and 16.

We fitted linear encoding models on the sources on AC and STG. TRF were computed for each model, subject and source using the Eelbrain toolbox[97]. For each model, corresponding speech features were shifted by T lags between −50 ms and 700 ms from the onset of each phoneme. The time window was set to 700 ms, as we did not expect neural responses to acoustic edges or phonemes to extend beyond this duration. Previous studies employing the TRF method for encoding acoustic edges and phoneme features used time windows ranging from 500 ms to 1000 ms but reported no significant effects beyond 700 ms[1,4,5,7,24,38,41]. With 50 ms wide Hamming windows at 100 Hz sampling rate that yields $T = 75$ time points. MEG response at time $t$ $\{y_i(t_n)\}_{j=1}^N$ ($N = 169$ virtual current source, $i$: subject number, $t_n$: time point) was predicted by convolving the TRF with predictor features shifted by $T$ time delays $\{x_f(t_n - \tau_k)\}_{f=1}^F$ (F: number of speech features in the model). $\beta_{iif}(\tau_k)$ is the TRF of $i$th subject, $j$th source point, $f$th speech features at $k$th latency.

$$y_i^j(t_n) = \sum_{f=1}^F \sum_{k=1}^T \beta_{iif}(\tau_k) x_f(t_n - \tau_k)$$

All predictors and MEG signals were normalized by subtracting the mean and dividing by the standard deviation. To prevent edge artifacts, they are zero padded until −50 ms before the onset and 700 ms after the offset. To estimate TRFs, boosting algorithm of Eelbrain toolbox was used to minimize the *l2 error* between the MEG signal and predicted signal using a fivefold cross-validation procedure. We used the early stopping from the toolbox. It uses a validation set, which is distinct from the test set to stop training when the error starts to increase to prevent overfitting[97].

## Statistics and reproducibility

Model accuracy was calculated as the proportion of variance in the neural signal explained by the model. To evaluate if acoustic edges and phoneme features significantly contributed to the model accuracy, accuracy differences of each subject between the full model which has all features and the model without the feature of interest were first averaged over sources and then tested against zero with a *t* test. To test effect of condition (sentences vs word lists or word lists vs syllables) on the averaged accuracy improvement over sources by acoustic edges and phoneme features, a LME was fitted using the lmer function in the lme4 package for R with factors condition and feature (acoustic edges and phoneme features) and a random effect for subjects. Random slope for condition and feature did not always converge, so we added them when they did. It is indicated on the LME formula for each analysis. We reported the output of the lmerTest package's anova() function for the fitted model to obtain Type III F-statistics with Satterthwaite-approximated degrees of freedom and demonstrate the main and interaction effects

To investigate the effect of condition on TRF weights, average of the absolute values of weights over sources and each time lag were compared by using cluster permutation test across time with 30,000 permutations[104] implemented on mne- python (version 0.23.1). We did not include the weights on the first and last 20 ms time interval to eliminate edge artifacts.

Given that peak times of TRF weights were not normally distributed we conducted an ANOVA using ART with the artlm() function from the ARTool package in R.

### Reporting summary

Further information on research design is available in the Nature Portfolio Reporting Summary linked to this article.

## Data availability

Processed MEG data and Source data underlying the figures in this paper are available in the Radboud University Repository database and are openly available with the identifier https://doi.org/10.34973/jedg-5009.

## Code availability

The code supporting the findings of this study is available on the GitHub repository at https://github.com/tezcanf/Phoneme_encoding_sentences_words_syllables.

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

## Acknowledgements

We Laurel Ellen Brehm for their help on the linear mixed effect models. AEM was supported by a Lise Meitner Research Group "Language and Computation in Neural Systems" from the Max Planck Society, and by the Netherlands Organization for Scientific Research (NWO) VIDI grant 016.Vidi.188.029 and Aspasia grant 015.014.013, and by an ERC Consolidator grant from the European Research Council (DYNALANG; ERC-2024-COG-101170162). STO was additionally supported by the European Research Council (ERC-2023-STG; 101116685) and by the Dutch Research Council (NWO; Starter Grant awarded to starting assistant professors).

## Author contributions

Conceptualization: F.T., A.E.M.; Data Collection: F.T., S.T.O, F.B., N.T.R; Methodology and Software: F.T., A.E.M.; Formal Analysis: F.T., S.T.O, A.E.M.; Writing- original draft: F.T.; Reviewing and Editing: all authors; Supervision, Resources, and Funding Acquisition: A.E.M.

## Funding

## Competing interests

The authors declare no competing interests.
