## [Transparent Peer Review file · Communications Biology]

Linguistic structure and language familiarity sharpen phoneme encoding in the brain

Corresponding Author: Ms Filiz Tezcan

Version 0:

Reviewer comments:

Reviewer #1

(Remarks to the Author)

Tezcan and colleagues used several MEG datasets from participants of diverse linguistic backgrounds (Dutch, Mandarin, Turkish speakers) to compare source localised ROI-specific differences in the cortical tracking (mTRF) of acoustic and phonological features in different contexts: isolated sentence processing, word (lexical) and pseudoword (non-lexical) lists. Authors showed that in some cases phoneme encoding was stronger in sentence contexts and (more consistently across languages) in word lists. The second aim of the study was to identify the effects of language familiarity on acoustic edge and phonological feature tracking in words and pseudowords (randomised syllables). Authors showed that exposure to a given language even without the ability to comprehend it modulates the ability to track phonological information in that language. While this study offers novel insights and important and understudied cross-linguistic comparisons, some of conclusions drawn need to be strengthened by appropriate analyses and statistical tests. Please see my comments for each section of the study.

Introduction:

1. A fairly minor point. The introduction opens with tracking defined as neural signal phase/amplitude alignment to stimulus features, with emphasis on the theta/delta bands. However, the majority of the literature cited is not band-specific with encoding/decoding of linguistic/acoustic features done on the broad-spectrum signal. Delta/theta distinction is not further referenced or discussed in the manuscript but this opening does set up this expectation for the reader.

1. Again, minor point. Lines 96 – 105 are not very clear, can be broken down into separate sentences.

2. The concept of “acoustic edges” is still not as popularised/prominent in recent literature so the reader would benefit from a clearer definition in the introduction, preferably with emphasis on how this is distinct from envelope tracking.

Methods

1. ROI selection for this study was motivated by the peaks of the Word Entropy by language interaction from the Tezcan et al., 2023 study. However, models, contrasts and participant groups tested in the present study go beyond re-testing previously reported effects/comparisons. Therefore, it is important to verify that the ROIs selected for source localisation still capture the peaks of responses for the relevant models across these multiple datasets.

2. One of the key methodological issues which impacts interpretability of the findings is not testing effects of Phoneme onsets vs Phoneme surprisal/ entropy independently during model comparisons stage. Please see my motivation for this and how this impacts interpretation in comment 2 of the Discussion section.

3. To test accuracy improvement for a given feature encoding, authors compared encoding accuracy of the full model to that of the reduced model which excluded the feature/s of interest. Another approach in the literature is to instead include into the “test” model the permuted version of that feature, with motivation of matching the number of regressors across models. It would be useful to verify (perhaps in a supplementary analysis) that author’s choice of accuracy improvement test (exclusion vs permutation) did not significantly impact their findings.

Discussion:

1. Lines 387-389. “Our findings indicate that phoneme feature weights within sentences were greater compared to word lists in a comprehended language, also an enhancement of phoneme feature weights”. This statement does not describe the results obtained with sufficient accuracy as the presented findings are not always consistent across encoding accuracy vs weights, as well as across languages (Dutch vs Turkish). First, authors would need to comment as to why the differences in phoneme encoding in sentence vs word lists are only consistency present in weights and not in accuracy (no significant

differences in accuracy improvements r-squared values either in Dutch or Turkish). Authors expand on this in lines 428-436 but in insufficient detail. Second, in the weights themselves, in Turkish we see the first significant cluster (Fig 2 D, bottom) showing a reverse effect – with greater weights for phonemes in world lists – which is inconsistent with author’s original hypotheses. In contrast, such effects appear to be much more consistent when comparing phoneme encoding in words vs syllables (differences in the same direction words>syllables present in accuracies and weights).

2. When authors discuss tracking of phoneme features, it is important to make a distinction between tracking the features themselves (i.e. cognitive computation of the content of the phoneme from the spectral characteristics / elements) vs tracking the statistical/distributional properties of the phonological content within words of the given language. These things seem to be conflated in the analysis and the discussion. Arguably the 2 out of 3 models that test for “phoneme tracking”, i.e. Phoneme surprisal and Entropy, are testing for the statistical/distributional properties of phonemes within words, while Phoneme onset tests for overall sensitivity to presence of the phoneme. I would argue strongly against putting them into a single test and would like to see the “accuracy improvements” for Phoneme onsets vs Phoneme surprisal and Entropy separately.

To improve the interpretability of the models further, I would recommend including models based on phoneme features vectors or spectral bands, such models would test directly neural encoding of phonological content.

Testing such models has important consequences for the discussion/conclusions drawn. For instance, consistent with literature discussed in the introduction, one could expect that in connected sentences (even in the absence of larger discourse context) local semantic predictions/constraints would diminish processing of phoneme content, but may boost processing of the statistical/distributional properties of the phonological elements. Such analyses would more explicitly test some interpretations made by authors later in the discussion (e.g. 454-460).

3. Lines 400-411. To support the statements/conclusions based on contrasting phoneme/edge encodings in familiar vs unfamiliar uncomprehended speech, authors would need to directly compare encoding accuracy and weights across different participant groups with appropriate between-groups tests. The same holds for conclusions drawn from comparisons of phoneme/edge encoding in words vs random syllables (pseudowords) across different languages (Dutch, Turkish, Mandarin; lines 488 - 500).

4. Lines 403-406. To support the statements/conclusions about peak latency differences, authors would need to conduct formal tests contrasting peak latencies in different conditions within the relevant participant groups (c.f. Broderick et al. 2021). Same for statements in lines 518-526 when contrasting latencies of encoding in weights for familiar vs native language in Mandarin speakers.

Reviewer #2

(Remarks to the Author)

REVIEW

SUMMARY

This study was looking at several previously published datasets in order to examine new questions. The authors examines how different sources of predictive information modulate acoustic encoding. They specifically looked at linguistic structure (lexical knowledge) and linguistic familiarity (experience with a language). Participants were speakers of Dutch, Mandarin, and Turkish, and their MEG was measured. MEG responses to acoustic edges and phoneme boundaries were examined, and the results showed primarily that experience with a language (either by familiarity or through lexical-level information) modulates phoneme-level processing. Processing of acoustics (e.g. edges) was not modulated by experiential differences in most analyses.

MAJOR CONCERNS

I enjoyed reading this paper, and I found it added nicely to the existing literature. However, I did have some major concerns that I hope the authors could address in a revision.

First, I think the introduction needs to be expanded and clarified. The authors understandably focus on studies with MEG that use a similar continuous-in-time analysis to natural speech stimuli, but there are also several other theoretical models of speech aside from Marslen-Wilson that could help frame their rationale for a broader audience.

Similarly, in the Discussion, the authors say:

“Our analyses suggest that statistical information can be obtained either from the mental lexicon, in the case of native language comprehension, or from experience with a spoken language, in the absence of language comprehension. Even short-term repeated exposure to an uncomprehended spoken language, as in the experimental setups for the Dutch and Mandarin Chinese languages, can increase the tracking of phoneme features by the brain; yet the time dynamics of phoneme feature-tracking show distinct profiles in a native versus nonnative language.”

I would love to hear a more nuanced discussion of this from the authors. They do a good job of explaining what they found in their results, but I would love to hear more of these types of bigger picture theoretical implications. Some literature that to me seems particularly relevant and might be worth looking into is the lexicalization of nonwords—where exposure to the words can cause them to sufficiently lexicalize and behave in many ways like words, even without semantic meaning. The authors mention exposure to a language in terms of statistical regularities, but I wonder if this nonword lexicalization could be another source of predictive information that could affect acoustic-phonemic processing.

Finally, in the Methods and throughout the Results, I would like to hear more details about the statistical models, and I would appreciate it if the authors could report their F-ratios and t-statistics in the typical convention, with degrees of freedom included:

$F(df_{\text{between}}, df_{\text{within}}) = F\text{-ratio}, p = p\text{-value}$

$t(df) = t\text{-value}, p = p\text{-value}$

Also, as a nonexpert on acoustic edges, I did not fully understand what this really refers to in terms of the features of the acoustic signal. Could the authors please clarify early on in the manuscript and maybe further elucidate with an updated visualization in Figure 1A?

MINOR CONCERNS

In Introduction, the authors may also consider broadening and adding some discussion of why acoustic features are less encoded with higher proficiency, particularly in contrast with the idea of perceptual tuning?
Line 168, “more words in a sentence”—Was this also a statistically significant difference? Please report in that same area. Moreover, was the number of phonemes or acoustic edges per sentence different?
Line 175, This is picky but should LLM be abbreviated as LME, (to me, LLM = large language model)
Line 203, 700 ms seems like a short epoch, why was this chosen as such?
Line 204, Please add more details about cluster-based permutation?
Line 235, Authors can just use LME again since they already introduced it
Line 246 and elsewhere, If you are reporting LME output, why do you have F ratios? Should these be beta coefficients? What package was used to run the LMEs? Was it the LME4 package in R?
Line 303, typo, “listening to Mandarin Chinese stimuli”
Line 331 & 403, typo, “uncomprehended”
Line 412-3, typo, the period should go after the citations

Reviewer #3

(Remarks to the Author)

The study by Tezcan et al. analyzes three datasets to examine how both linguistic structure and experience-based familiarity shape neural tracking of acoustic and phonemic features. Across datasets, the authors consistently find that phonemic tracking becomes stronger as linguistic processing deepens—whether through the construction of larger linguistic units, greater language proficiency, or increased familiarity with the language.

Overall, the paper offers a comprehensive view of how acoustic and phonemic responses are modulated by linguistic structure and familiarity. I enjoyed the abstract and the opening of the introduction; both adopt a broad, insightful perspective and clearly summarize the compelling main results. My main concerns relate to the organization and presentation of the results, with a view to improving readability and helping readers extract the key findings easier.

1. The inconsistent results between reconstruction accuracy and TRF weights. First, the reconstruction accuracy is not formally defined in this manuscript. If I understand correctly, the reconstruction accuracy represents the R square value between the predicted and measured MEG signal? If so, the ‘reconstruction’ should be called as ‘predictive/prediction’ as reconstruction usually refers to reconstructing speech features (e.g., the envelope) from neural signals

The reconstruction accuracy generally shows a consistent pattern as TRF weights, which means if the neural response to a feature is larger, then the accuracy in predicting the response using this feature would also be larger. As summarized in Fig. 6, such inconsistency often appears in the feature of acoustic edges. Drawing a conclusion given such inconsistency may not fully convince readers. I suggested the authors offer a plausible explanation for the inconsistency or reconsider this claim.

Additionally, from my perspective, the main results of this study are phoneme encoding, which are overall consistent in terms of prediction accuracy and TRF weights, and are interesting enough. I am uncertain if the claim about acoustic responses enhances the current study a lot or just adds confusion.

2. Also, how the basic acoustic response is modulated by structural building or by language familiarity has been studied in many relevant literatures, for either natural speech (Decruy et al., 2019, 2020; Reetzke et al., 2021; Zou et al., 2019) and isochronous speech (Chen et al., 2020), which consistently show that basic acoustic responses decrease with higher language proficiency. These findings are not consistent with the results of the current study. More detailed discussion is needed.

3. Some comparisons of TRF weights appear a bit problematic. For example, Figures 2C and 4D (which should be labeled 4E) seem to represent a baseline difference between the two waveforms, resulting in significant differences across all time lags, even before zero. Authors should verify whether these significant differences reflect meaningful effects or arise from baseline differences, as the power of TRF weights could be affected by the noise levels and samples for the average

4. Does Figure 5A compare Chinese-speaking participants listening to Mandarin versus Dutch speech? If so, differing speech materials complicate interpretation. A more appropriate comparison might involve participants with different language familiarity listening to identical materials—for example, comparing Chinese and Dutch participants both listening to Dutch.

5. As this study is conducted based on multiple datasets, including multiple languages and multiple structures, it gives a comprehensive view of how acoustic and phonemic encoding are affected by structure building and language familiarity (which I admire and appreciate). However, it also brings the difficulty of figuring out exactly what stimuli were used in each dataset and what the corresponding tasks were. I suggest that the authors reorganize Figure 1 or include a separate figure that illustrates the structure of the stimuli, the task, and the participants, so that readers don’t need to refer to the Methods section or references to find this information.

6. The discussion is currently challenging to navigate. To improve readability, the authors may consider structuring it with

subheadings.

References

- Chen, Y., Jin, P., & Ding, N. (2020). The influence of linguistic information on cortical tracking of words. *Neuropsychologia*, 148, 107640. <https://doi.org/10.1016/j.neuropsychologia.2020.107640>
- Decruey, L., Vanthornhout, J., & Francart, T. (2019). Evidence for enhanced neural tracking of the speech envelope underlying age-related speech-in-noise difficulties. *Journal of Neurophysiology*, 122(2), 601–615. <https://doi.org/10.1152/jn.00687.2018>
- Decruey, L., Vanthornhout, J., & Francart, T. (2020). Hearing impairment is associated with enhanced neural tracking of the speech envelope. *Hearing Research*, 393, 107961. <https://doi.org/10.1016/j.heares.2020.107961>
- Reetzke, R., Gnanateja, G. N., & Chandrasekaran, B. (2021). Neural tracking of the speech envelope is differentially modulated by attention and language experience. *Brain and Language*, 213, 104891. <https://doi.org/10.1016/j.bandl.2020.104891>
- Zou, J., Feng, J., Xu, T., Jin, P., Luo, C., Zhang, J., Pan, X., Chen, F., Zheng, J., & Ding, N. (2019). Auditory and language contributions to neural encoding of speech features in noisy environments. *NeuroImage*, 192, 66–75. <https://doi.org/10.1016/j.neuroimage.2019.02.047>

Version 1:

Reviewer comments:

Reviewer #1

(Remarks to the Author)

I thank the authors for addressing my initial concerns and running additional analyses to further substantiate arguments they make in the discussion. I do not have any further concerns.

Reviewer #3

(Remarks to the Author)

The authors have addressed my previous concerns. I recommend this study for publication in *Communications Biology*.

Dear Dr. Bessieres,

We appreciate the chance to submit a revised version of our manuscript titled "Linguistic structure and language familiarity sharpen phoneme encoding in the brain." We have benefitted from the valuable feedback from you and the Reviewers, and have carefully addressed each concern. We conducted a series of new analyses, which support our claims, in addition to addressing Reviewer concerns, and we have thoroughly reworked the manuscript to address Reviewer comments.

In the responses to Reviewers' comments, the revised or added sections are highlighted in red.

In addition to the comment from Reviewers, df values for Mandarin and Dutch participants were corrected. Pairwise comparison tables of LMEs were also added.

Reviewers' comments:

Reviewer #1 (Remarks to the Author):

Tezcan and colleagues used several MEG datasets from participants of diverse linguistic backgrounds (Dutch, Mandarin, Turkish speakers) to compare source localised ROI-specific differences in the cortical tracking (mTRF) of acoustic and phonological features in different contexts: isolated sentence processing, word (lexical) and pseudoword (non-lexical) lists. Authors showed that in some cases phoneme encoding was stronger in sentence contexts and (more consistently across languages) in word lists. The second aim of the study was to identify the effects of language familiarity on acoustic edge and phonological feature tracking in words and pseudowords (randomised syllables). Authors showed that exposure to a given language even without the ability to comprehend it modulates the ability to track phonological information in that language. While this study offers novel insights and important and understudied cross-linguistic comparisons, some of conclusions drawn need to be strengthened by appropriate analyses and statistical tests. Please see my comments for each section of the study.

Introduction:

1. A fairly minor point. The introduction opens with tracking defined as neural signal phase/amplitude alignment to stimulus features, with emphasis on the theta/delta bands. However, the majority of the literature cited is not band-specific with encoding/decoding of linguistic/acoustic features done on the broad-spectrum signal. Delta/theta distinction is not further referenced or discussed in the manuscript but this opening does set up this expectation for the reader.

We thank the reviewer for this valuable comment. We have revised the Introduction to clarify the rationale for emphasizing the delta and theta bands and have added relevant references supporting this selection.

“In recent years, the investigation of neural tracking in the delta and theta bands, encompassing the alignment of neural activity phases with acoustic and linguistic features of speech signals, has become an integral component of contemporary theories in speech and language processing. It’s been shown that slow neural activity at delta/theta band recorded by MEG/EEG synchronizes with speech stimulus (Ahissar et al. 2001; Aiken and Picton 2008; Luo and Poeppel 2007; Cogan and Poeppel, 2011; Ding and Simon, 2012; Peelle, Gross and Davis, 2013; Gross et al, 2013; Doelling et al, 2014; Di Liberto et al., 2015). Alignment of the phase of the slow neural activity with sensory input is posited as a canonical mechanism influencing temporal aspects of speech perception (Giraud and Poeppel, 2012). Because linguistic information unfolds across timescales spanning both delta and theta ranges—though we do not suggest a strict one-to-one mapping—linguistic representations relevant for theories of neural encoding are likely reflected in modulations across these frequencies. Accordingly, analyzing the broadband signal captures the essential neural responses without imposing biases or

assuming an unsubstantiated correspondence between specific linguistic processes and particular frequency bands.

1. Again, minor point. Lines 96 – 105 are not very clear, can be broken down into separate sentences.

It is revised as follows:

“A strong test of how linguistic structure influences phoneme and acoustic edge encoding would be examination whether higher-level linguistic patterns enhance neural encoding even without naturalistic discourse context. This would reveal how local linguistic structure modulates neural responses to phonemes and acoustic edges independently of broader conversational meaning. A related question concerns the role of language exposure versus comprehension. Does statistical familiarity with a language's speech patterns, gained through daily exposure without understanding, such as living in a country where once does not speak the local language, affect the neural encoding of acoustic and/or phoneme features in a similar or different way to knowledge of linguistic structure.”

2. The concept of “acoustic edges” is still not as popularised/prominent in recent literature so the reader would benefit from a clearer definition in the introduction, preferably with emphasis on how this is distinct from envelope tracking.

We thank the reviewer for this helpful suggestion. To address this point, we have added a paragraph in the Introduction that more clearly defines “acoustic edges” and distinguishes them from envelope tracking:

Linguistic features of speech can be correlated with the acoustic features, and for example the neural tracking of phonemic features can also be explained by the tracking of rapid changes in acoustic envelope (e.g., acoustic edges; Daube et al., 2019; Doelling et al. 2014; Oganian et al. 2023). The acoustic envelope captures slow amplitude changes in speech signals, with its temporal patterns reflecting the timescales of syllables and words. Phoneme onsets correspond to acoustic onsets, termed “acoustic edges”, which are derived from half-wave rectified derivatives of the acoustic envelope (Brodbeck et al., 2018; Daube et al., 2019). To separate neural responses driven by acoustic versus linguistic information at the phoneme level, we incorporated acoustic edge features (an 8 logarithmically spaced frequency band acoustic onset spectrogram, which models sudden changes in the gammatone spectrogram of the audio signal using an acoustic edge detection model) alongside an auditory spectrogram with 8-band that represents the acoustic envelope (Fig 1A, See Methods).

Methods:

Acoustic Features: Acoustic features were generated using the Eelbrain toolbox (Brodbeck, 2023). This included an 8-band gammatone spectrogram and an 8-band acoustic onset spectrogram, which models sudden changes in the gammatone spectrogram of the audio signal using an acoustic edge detection model (Fishbach et al., 2001). Both spectrograms covered frequencies from 20 to 5000 Hz in equivalent rectangular bandwidth (ERB) space.

Methods

1. ROI selection for this study was motivated by the peaks of the Word Entropy by language interaction from the Tezcan et al., 2023 study. However, models, contrasts and participant groups tested in the present study go beyond re-testing previously reported effects/comparisons. Therefore, it is important to verify that the ROIs selected for source localisation still capture the peaks of responses for the relevant models across these multiple datasets.

We thank the reviewer for raising this important point about ROI selection. The ROI selection in our study was motivated not only by the findings of Tezcan et al. (2023), but also by a substantial body of prior research demonstrating that the auditory cortex (AC) and superior temporal gyrus (STG) encode

acoustic and phoneme-level features (e.g., Chang et al., 2010; Hamilton, Edwards, & Chang, 2018; Brodbeck et al., 2018; Daube et al., 2019; Donhauser & Baillet, 2020; Brodbeck et al., 2022; Leonard et al., 2024; Kim et al, 2024).

We restricted our analyses to these ROIs primarily for computational efficiency. The boosting algorithm we employed is considerably more computationally demanding than ridge regression, and with data from 79 participants, each model required over 20 hours to run for whole brain analysis. Because we trained four models per participant to incrementally assess accuracy improvements across features of interest, this amounted to 760 models in total. The ROI (169 source points) is substantially smaller than the whole-brain model (5,124 source points), reducing computation time by approximately 40 minutes per model while still capturing the primary auditory and phoneme-level processing regions.

To confirm that our ROI selection was appropriate, we additionally conducted a whole-brain analysis on the first Dutch dataset. This analysis showed that model accuracy for both acoustic edge and phoneme features increased significantly around the AC and STG regions similar to previous studies. We have revised the corresponding paragraph in the manuscript to clarify our rationale for ROI selection and to include this verification.

Figure 1. Accuracy improvement by acoustic edges and phoneme features for sentences and words condition in Dataset 1. Color bar indicates t values of permutation cluster t test.

In our earlier study (Tezcan et al., 2023), we found that language comprehension and contextual information modulated the encoding of acoustic edges and phoneme features, with the strongest effects observed in the auditory cortex (AC) and superior temporal gyrus (STG). Other studies also consistently showed that these regions encode acoustic edges and phoneme features (Chang et al., 2010; Hamilton, Edwards, and Chang, 2018; Brodbeck et al., 2018; Daube et al., 2019; Donhauser and Baillet, 2020; Brodbeck et al., 2022; Leonard et al., 2024; Kim et al, 2024). Additionally, our previous study showed that averaged responses to all phoneme features (onset, surprisal, and entropy) were more consistent across story segments than individual feature responses. This likely reflects the complementary nature of these features: phoneme onset captures time-invariant categorization responses, while surprisal and entropy, derived from cohort model statistics (See Methods), reflect contextual modulation at the word level. Since we aimed to examine encoding differences between sensory acoustic information and abstract linguistic units, and phoneme-level statistical features also capture information about categorized phonemes, we analyzed averaged responses to all phoneme features in the AC and STG. Instead of running separate whole-brain analyses for each feature, this approach reduced computational demands when analyzing three different datasets with nine different contrasts. This allowed us to investigate how linguistic structure modulates acoustic edges and

phoneme features by comparing both model accuracy improvements and model weights, thereby examining the temporal dynamics of feature-specific neural contributions.

2. *One of the key methodological issues which impacts interpretability of the findings is not testing effects of Phoneme onsets vs Phoneme surprisal/ entropy independently during model comparisons stage. Please see my motivation for this and how this impacts interpretation in comment 2 of the Discussion section.*

We appreciate the reviewer's thoughtful and constructive feedback. We replied this comment on the comments related to the Discussion.

3. *To test accuracy improvement for a given feature encoding, authors compared encoding accuracy of the full model to that of the reduced model which excluded the feature/s of interest. Another approach in the literature is to instead include into to the "test" model the permuted version of that feature, with motivation of matching the number of regressors across models. It would be useful to verify (perhaps in a supplementary analysis) that author's choice of accuracy improvement test (exclusion vs permutation) did not significantly impact their findings.*

As we aimed to compare continuous features (acoustic edges) and sparse features (phoneme features), we used boosting algorithm implemented in Eelbrain toolbox (David et al., 2007; David and Shamma, 2013; Brodbeck et al., 2021) instead using ridge regression. While boosting algorithm is computationally more costly than ridge regression, we chose boosting due to fundamental differences in how these methods handle feature selection and regularization. Ridge regression applies L2 regularization uniformly across all features, which may be suboptimal when features differ substantially in sparsity. This uniform penalization can be particularly problematic when combining sparse features (like phoneme onset, surprisal and entropy) with continuous features (like envelope and acoustic edges).

Boosting algorithms, in contrast, perform implicit feature selection by iteratively selecting the most informative features at each iteration step. It starts with setting all weights to zero then in each iteration gradually increases/decreases the weight of each feature to optimize the prediction accuracy. The algorithm naturally handles features with different characteristics by updating the weights of the feature more that contribute more to prediction accuracy. It handles overfitting through early stopping mechanisms that halt updating the weight of a specific feature when validation performance plateaus, whereas with ridge regression it is handled by comparison with surrogate (shuffled) features (See Brodbeck et al., 2021). Because of this difference, adding additional features that do not explain the neural response does not increase explained variance. We demonstrated this by comparing the accuracy improvement attributed to phoneme surprisal and entropy using two approaches: (1) the incremental method used in our study, where the feature of interest is added to the model and the variance explained by the base model is subtracted from that of the full model, and (2) a shuffling method, where only the values of the feature of interest are shuffled (while preserving their time points) and the model with the shuffled feature plus all other features serves as the base model. To compare accuracy improvements computed with the incremental versus shuffling methods for both the word and sentence conditions in Dataset 1, we fit a linear mixed-effects model (LME). The model without the method \times condition interaction, but including random slopes for subjects and condition, showed a lower Bayesian Information Criterion (BIC) than the model with the interaction (BIC -995.95 vs. -787.12 ; $\Delta\chi^2 = 227.15$, $p < 0.0001$). We observed a significant main effect of method, with the shuffling method yielding larger accuracy improvements than the incremental method ($F(1,18) = 5.17$, $p = 0.035$). Pairwise comparisons using the *lsmeans* function in R (Lenth, 2016) did not reveal any significant differences in accuracy improvement (see Fig. 3 and Table 2). Overall, this comparison indicates that the incremental method is slightly more conservative than the shuffling method.

Figure 3. Accuracy improvement by phoneme surp./entr features calculated with incremental and shuffling methods in Dataset 1.

Table 2. LME results of phoneme surp./entr features for incremental and shuffling method comparison in Dataset 1.

	Sum Sq	Mean Sq	NumDF	DenDF	F value	Pr(>F)
condition	4.43E-09	4.43E-09	1	18.006	5.1772	3.53E-02
feature	4.03E-09	4.03E-09	1	37.002	4.7119	3.64E-02

Discussion:

1. Lines 387-389. *“Our findings indicate that phoneme feature weights within sentences were greater compared to word lists in a comprehended language, also an enhancement of phoneme feature weights”. This statement does not describe the results obtained with sufficient accuracy as the presented findings are not always consistent across encoding accuracy vs weights, as well as across languages (Dutch vs Turkish). First, authors would need to comment as to why the differences in phoneme encoding in sentence vs world lists are only consistency present in weights and not in accuracy (no significant differences in accuracy improvements r-squared values either in Dutch or Turkish). Authors expand on this in lines 428-436 but in insufficient detail.*

We thank the reviewer for this valuable suggestion. In response, we have expanded our introduction to clarify what the model accuracy and weights represent in this analysis. While we acknowledge that drawing definitive conclusions from these results remains challenging, we have added a detailed discussion exploring potential explanations for the differential modulation of accuracy and weights by sentence- versus word-level factors.

Added in Introduction: *“We compared the improvement in model accuracy attributed to the inclusion of acoustic and phoneme features, defined as the increase in the proportion of neural signal variance explained by the model when these features were added. This measure reflects how well the real neural signal aligns with the model’s predicted signal in terms of both phase and power. In addition, we compared the model weights of the filters learned by the model that represent the neural responses evoked by specific features. This measure enables us to assess differences in the amplitude of neural responses contributing to the predicted signal.”*

Added in Discussion: “The differential patterns we observe where word-syllable contrasts affect both accuracy and weights, while sentence-word contrasts affect only weights likely reflect distinct underlying neural mechanisms. When processing random syllables versus words, the brain might be recruiting distinct cortical circuits with different oscillatory signatures, resulting in differences in both phase alignment (captured by accuracy) and power modulation (reflected in weights). In contrast, words presented in isolation versus within sentences might be utilizing the same network, but with altered temporal dynamics. The sentence structure might modulate how this network processes word-level information adjusting the timing and strength of responses rather than recruiting entirely different circuits. “

Second, in the weights themselves, in Turkish we see the first significant cluster (Fig 2 D, bottom) showing a reverse effect – with greater weights for phonemes in word lists – which is inconsistent with author’s original hypotheses. In contrast, such effects appear to be much more consistent when comparing phoneme encoding in words vs syllables (differences in the same direction words>syllables present in accuracies and weights).

We thank the reviewer for highlighting this important point. Although we reported the second cluster in the Results section, we inadvertently omitted it from the Discussion. We have now added the following paragraph to address this in the revised manuscript.

In addition, we observed a reduction in phoneme-feature weights during the early time interval in the Turkish dataset, an effect not present in the Dutch dataset. A key distinction between these two datasets is that, in the Turkish dataset, the stimuli consist of isochronous words. In the sentences condition, these words form repeating sentence structures, whereas in the words condition they do not. This pattern may parallel the early-interval suppression of acoustic-edge responses previously reported for words compared to syllables, which has been attributed to perfectly predictable timing and/or stimulus repetition. By analogy, the repetition of sentence structure or the highly predictable temporal patterning in the Turkish stimuli may suppress phoneme-related responses at early latencies. However, this interpretation remains speculative, and future studies are needed to directly test whether temporal regularity or structural repetition modulates phoneme-level encoding in this way.

2. When authors discuss tracking of phoneme features, it is important to make a distinction between tracking the features themselves (i.e. cognitive computation of the content of the phoneme from the spectral characteristics / elements) vs tracking the statistical/distributional properties of the phonological content within words of the given language. These things seem to be conflated in the analysis and the discussion. Arguably the 2 out of 3 models that test for “phoneme tracking”, i.e. Phoneme surprisal and Entropy, are testing for the statistical/distributional properties of phonemes within words, while Phoneme onset tests for overall sensitivity to presence of the phoneme. I would argue strongly against putting them into a single test and would like to see the “accuracy improvements” for Phoneme onsets vs Phoneme surprisal and Entropy separately. To improve the interpretability of the models further, I would recommend including models based on phoneme features vectors or spectral bands, such models would test directly neural encoding of phonological content. Testing such models has important consequences for the discussion/conclusions drawn. For instance, consistent with literature discussed in the introduction, one could expect that in connected sentences (even in the absence of larger discourse context) local semantic predictions/constraints would diminish processing of phoneme content, but may boost processing of the statistical/distributional properties of the phonological elements. Such analyses would more explicitly test some interpretations made by authors later in the discussion (e.g. 454-460).

We appreciate the reviewer’s feedback. Our study aimed to investigate how linguistic structure, from syllables to sentences, and associated statistical information influence both acoustic and linguistic

encoding at the phoneme level. By jointly examining phoneme surprisal, entropy, and onsets, we sought to understand how various aspects of categorical phoneme-related neural responses and acoustic information that guide phoneme categorization are shaped by higher-level linguistic structure.

We agree with the reviewer’s comment that surprisal and entropy capture the statistical and distributional properties of phonemes, while phoneme onsets reflect sensitivity to phoneme occurrence independent of distributional predictability. It is important to note that tracking surprisal and entropy inherently requires phoneme categorization, which involves abstracting from the acoustic dimensions (spectral and temporal cues) that distinguish phoneme categories. Conversely, as TRF modelling capture temporally invariant response to the model features, modeling phoneme onsets alone would assume temporally uniform categorical responses across phonemes, overlooking the systematic variation in phoneme-evoked responses according to their statistical properties. Following the reviewer’s recommendation, we have conducted separate analyses for all datasets (sentence vs. word-list, and words vs. syllables) examining phoneme onset and surprisal/entropy independently. The findings align with our expectation; word- and sentence-level structure primarily modulate surprisal and entropy, with no effects on phoneme onsets. Additionally, our results demonstrate that phoneme onsets and surprisal/entropy each make independent contributions to model accuracy.

Regarding the incorporation of phonological feature vectors or spectral-band representations; our acoustic model already includes spectral bands and spectral power changes (acoustic edges), which provide the foundational information for phoneme categorization. We acknowledge that including explicit phonological feature-based models would offer valuable insights into which acoustic dimensions contribute most, especially to differences between native and non-native language processing. We consider this an important avenue for future research, as the current study focused specifically on comparing differences between acoustic and abstract categorical information.

Figure 5. Accuracy improvement for Dutch (Dataset 1) and Turkish (Dataset 3) stimuli **A, B**

Accuracy improvement averaged over all sources on both right and left AC and STG for sentences and word lists in Dutch and Turkish stimuli, respectively. Box edges indicate the standard error of the mean. Black line shows the mean. (**** < 0.0001, *** < 0.001, ** < 0.01, * < 0.05)

Figure 6. Accuracy improvement **A** Mandarin-speaking participants listening to Mandarin Chinese stimuli, **B** Mandarin-speaking participants listening to Dutch stimuli, **C** Dutch-speaking participants listening to Mandarin Chinese stimuli, **D** Dutch-speaking participants listening to Dutch stimuli, **E** Turkish-speaking participants listening to Turkish stimuli. Accuracy improvement averaged over all

sources on both right and left AC and STG for words and syllables. Box edges indicate the standard error of the mean. Black line shows the mean. (**** <0.0001, *** <0.001, ** <0.01, * < 0.05)

Table 4. Accuracy improvement by phoneme onset and surprisal/entropy features in sentences vs words condition (Dataset1, Dutch stimuli)

Features	t-statistic	df	p-value
Sentences Phoneme Onset	6.94	18	< 0.0001
Sentences Phoneme Surprisal/Entropy	6.97	18	< 0.0001
Words Phoneme Onset	6.7	18	< 0.0001
Words Phoneme Surprisal/Entropy	6.7	18	< 0.0001

Table 5. LME results of phoneme onset and surprisal/entropy features in sentences vs words condition (Dataset1, Dutch stimuli)

	Sum Sq	Mean Sq	NumDF	DenDF	F value	Pr(>F)	
condition	4.87E-09	4.87E-09	1	17.996	4.7951	0.04196	*
feature	1.25E-08	1.25E-08	1	36.999	12.2931	1.21E-03	**

Table 6. Pairwise comparison of LME (Dataset1, Dutch stimuli)

contrast	estimate	SE	df	t.ratio	p.value
Sentences Phoneme Onset – Words Phoneme Onset	6.60E-04	3.02E-04	18	2.19	0.1639
Sentences Phoneme Onset – (Sentences Pho Surp/Ent)	-2.56E-05	7.31E-06	37	-3.506	0.0064
Sentences Phoneme Onset – (Words Pho Surp/Ent)	6.35E-04	3.02E-04	18	2.104	0.1894
Words Phoneme Onset – (Sentences Pho Surp/Ent)	-6.86E-04	3.02E-04	18	-2.274	0.1414
Words Phoneme Onset – (Words Pho Surp/Ent)	-2.56E-05	7.31E-06	37	-3.506	0.0064
(Sentences Pho Surp/Ent) – (Words Pho Surp/Ent)	6.60E-04	3.02E-04	18	2.19	0.1639

Table 7. Accuracy improvement by phoneme onset and surprisal/entropy features in sentences and words condition (Dataset3, Turkish stimuli)

Features	t-statistic	df	p-value
Sentences Phoneme Onset	3.44	29	< 0.0001
Sentences Phoneme Surprisal/Entropy	9.2	29	< 0.0001
Words Phoneme Onset	8	29	< 0.0001
Words Phoneme Surprisal/Entropy	9.07	29	< 0.0001

Table 8. LME results of phoneme onset and surprisal/entropy features in sentences vs words condition (Dataset3, Turkish stimuli)

	Sum Sq	Mean Sq	NumDF	DenDF	F value	Pr(>F)	
Condition	6.97E-08	6.97E-08	1	55.34	7.24	9.41E-03	**
Feature	1.15E-08	1.15E-08	1	29.00	1.19	2.84E-01	
Condition:Feature	1.13E-07	1.13E-07	1	58.00	11.69	1.16E-03	**

Table 9. Pairwise comparison of LME (Dataset3, Turkish stimuli sentences vs words)

contrast	estimate	SE	df	t.ratio	p.value
Sentences Phoneme Onset – Words Phoneme Onset	-1.25E-05	2.55E-05	58	-0.492	0.9606
Sentences Phoneme Onset – (Sentences Pho Surp/Ent)	-9.95E-05	3.94E-05	43.2	-2.528	0.0696
Sentences Phoneme Onset – (Words Pho Surp/Ent)	1.05E-05	4.01E-05	29	0.261	0.9936
Words Phoneme Onset – (Sentences Pho Surp/Ent)	-8.70E-05	3.87E-05	29	-2.246	0.1349
Words Phoneme Onset – (Words Pho Surp/Ent)	2.30E-05	3.94E-05	43.2	0.585	0.936
(Sentences Pho Surp/Ent) – (Words Pho Surp/Ent)	1.10E-04	2.55E-05	58	4.318	0.0004

Table 10. Accuracy improvement by phoneme onset and surprisal/entropy features in words and syllables condition (Dataset2, Mandarin Speaking participants)

Features	t-statistic.	Df	p-value
Dutch Stimuli Words Phoneme Onset	4.84	13	0.00032
Dutch Stimuli Words Phoneme Surprisal/Entropy	7.22	13	< 0.0001
Chinese Stimuli Words Phoneme Onset	3.03	13	0.00973
Chinese Stimuli Words Phoneme Surprisal/Entropy	4.79	13	0.00035
Dutch Stimuli Syllables Phoneme Onset	5.36	13	0.00013
Dutch Stimuli Syllables Phoneme Surprisal/Entropy	2.64	13	0.02035
Chinese Stimuli Syllables Phoneme Onset	2.63	13	0.02066
Chinese Stimuli Syllables Phoneme Surprisal/Entropy	3.74	13	0.00247

Table 11. LME results of phoneme onset and surprisal/entropy features in words vs syllables condition (Dataset2, Mandarin speaking Participants, Mandarin Chinese stimuli)

	Sum Sq	Mean Sq	NumDF	DenDF	F value	Pr(>F)
Condition	7.81E-06	7.81E-06	1	16.24	15.73	1.08E-03
Feature	1.50E-05	1.50E-05	1	39.00	30.29	2.53E-06
Condition:Feature	1.56E-05	1.56E-05	1	39.00	31.31	1.89E-06

Table 12. LME results of phoneme onset and surprisal/entropy features in words vs syllables condition (Dataset2, Mandarin speaking Participants, Dutch stimuli)

	Sum Sq	Mean Sq	NumDF	DenDF	F value	Pr(>F)
Condition	1.69E-05	1.69E-05	1	39.00	54.91	5.91E-09
Feature	1.46E-05	1.46E-05	1	39.00	47.34	3.13E-08
Condition:Feature	1.41E-05	1.41E-05	1	39.00	45.90	4.38E-08

Table 13. LME results of phoneme onset and surprisal/entropy features in words vs syllables condition (Dataset2, Mandarin speaking Participants, Mandarin Chinese stimuli)

	Sum Sq	Mean Sq	NumDF	DenDF	F value	Pr(>F)
Condition	7.81E-06	7.81E-06	1	16.24	15.73	1.08E-03
Feature	1.50E-05	1.50E-05	1	39.00	30.29	2.53E-06
Condition:Feature	1.56E-05	1.56E-05	1	39.00	31.31	1.89E-06

Table 14. Pairwise comparison of LME (Dataset2, Mandarin speaking Participants, Dutch stimuli)

contrast	estimate	SE	df	t.ratio	p.value
Syllables Phoneme Onset – Words Phoneme Onset	-9.42E-05	2.10E-04	39	-0.45	0.97
Syllables Phoneme Onset – Syllables Pho Surp/Ent	-1.57E-05	2.10E-04	39	-0.08	1.00
Syllables Phoneme Onset – Words Pho Surp/Ent	-2.12E-03	2.10E-04	39	-10.11	<.0001
Words Phoneme Onset – Syllables Pho Surp/Ent	7.85E-05	2.10E-04	39	0.37	0.98
Words Phoneme Onset – Words Pho Surp/Ent	-2.03E-03	2.10E-04	39	-9.66	<.0001
Syllables Pho Surp/Ent – Words Pho Surp/Ent	-2.10E-03	2.10E-04	39	-10.03	<.0001

Table 15. Pairwise comparison of LME (Dataset2, Mandarin speaking Participants, Mandarin Chinese stimuli)

contrast	estimate	SE	df	t.ratio	p.value
Syllables Phoneme Onset – Words Phoneme Onset	3.53E-05	3.18E-04	26.9	0.11	1.00
Syllables Phoneme Onset – Syllables Pho Surp/Ent	1.72E-05	2.66E-04	26	0.07	1.00
Syllables Phoneme Onset – Words Pho Surp/Ent	-2.06E-03	3.18E-04	26.9	-6.45	<.0001
Words Phoneme Onset – Syllables Pho Surp/Ent	-1.81E-05	3.18E-04	26.9	-0.06	1.00
Words Phoneme Onset – Words Pho Surp/Ent	-2.09E-03	2.66E-04	26	-7.85	<.0001
Syllables Pho Surp/Ent – Words Pho Surp/Ent	-2.07E-03	3.18E-04	26.9	-6.51	<.0001

Table 16. Accuracy improvement by phoneme onset and surprisal/entropy features in words and syllables condition (Dataset2, Dutch participants)

Features	t-statistic.	Df	p-value
Dutch Stimuli Words Phoneme Onset	5.51	14	< 0.0001
Dutch Stimuli Words Phoneme Surprisal/Entropy	3.36	14	0.00462
Chinese Stimuli Words Phoneme Onset	4.59	14	0.00042
Chinese Stimuli Words Phoneme Surprisal/Entropy	5.38	14	< 0.0001
Dutch Stimuli Syllables Phoneme Onset	4.68	14	0.00035
Dutch Stimuli Syllables Phoneme Surprisal/Entropy	2.04	14	0.06026
Chinese Stimuli Syllables Phoneme Onset	4.75	14	0.00031
Chinese Stimuli Syllables Phoneme Surprisal/Entropy	4.12	14	0.00103

Table 17. LME results of phoneme onset and surprisal/entropy features in words vs syllables condition (Dataset2, Dutch speaking Participants, Dutch stimuli)

	Sum Sq	Mean Sq	NumDF	DenDF	F value	Pr(>F)
Condition	2.32E-05	2.32E-05	1	42.00	11.00	1.89E-03
Feature	2.30E-05	2.30E-05	1	42.00	10.91	1.96E-03
Condition:Feature	2.37E-05	2.37E-05	1	42.00	11.21	1.72E-03

Table 18. LME results of phoneme onset and surprisal/entropy features in words vs syllables condition (Dataset2, Dutch speaking Participants, Mandarin Chinese stimuli)

	Sum Sq	Mean Sq	NumDF	DenDF	F value	Pr(>F)
Condition	7.81E-06	7.81E-06	1	16.24	15.73	1.08E-03
Feature	1.50E-05	1.50E-05	1	39.00	30.29	2.53E-06
Condition:Feature	1.56E-05	1.56E-05	1	39.00	31.31	1.89E-06

Table 19. Pairwise comparison of LME (Dataset2, Dutch speaking Participants, Mandarin Chinese stimuli)

contrast	estimate	SE	df	t.ratio	p.value
Syllables Phoneme Onset – Words Phoneme Onset	3.53E-05	3.18E-04	26.9	0.11	1.00
Syllables Phoneme Onset – Syllables Pho Surp/Ent	1.72E-05	2.66E-04	26	0.07	1.00
Syllables Phoneme Onset – Words Pho Surp/Ent	-2.06E-03	3.18E-04	26.9	-6.45	<.0001
Words Phoneme Onset – Syllables Pho Surp/Ent	-1.81E-05	3.18E-04	26.9	-0.06	1.00
Words Phoneme Onset – Words Pho Surp/Ent	-2.09E-03	2.66E-04	26	-7.85	<.0001
Syllables Pho Surp/Ent – Words Pho Surp/Ent	-2.07E-03	3.18E-04	26.9	-6.51	<.0001

Table 20. Pairwise comparison of LME (Dataset2, Dutch speaking Participants, Dutch stimuli)

contrast	estimate	SE	df	t.ratio	p.value
Syllables Phoneme Onset – Words Phoneme Onset	-9.42E-05	2.10E-04	39	-0.45	0.97
Syllables Phoneme Onset – Syllables Pho Surp/Ent	-1.57E-05	2.10E-04	39	-0.08	1.00
Syllables Phoneme Onset – Words Pho Surp/Ent	-2.12E-03	2.10E-04	39	-10.11	<.0001
Words Phoneme Onset – Syllables Pho Surp/Ent	7.85E-05	2.10E-04	39	0.37	0.98
Words Phoneme Onset – Words Pho Surp/Ent	-2.03E-03	2.10E-04	39	-9.66	<.0001
Syllables Pho Surp/Ent – Words Pho Surp/Ent	-2.10E-03	2.10E-04	39	-10.03	<.0001

Table 21. Accuracy improvement by phoneme onset and surprisal/entropy features in words vs syllables condition (Dataset3, Turkish stimuli)

Features	t-statistic	df	p-value
Sentences Phoneme Onset	8.26	29	< 0.0001
Sentences Pho Surp/Ent	8.48	29	< 0.0001
Words Phoneme Onset	6.47	29	< 0.0001
Words Pho Surp/Ent	8.33	29	< 0.0001

Table 22. LME results of phoneme onset and surprisal/entropy features in words vs syllables condition (Dataset3, Turkish stimuli)

	Sum Sq	Mean Sq	NumDF	DenDF	F value	Pr(>F)
Condition	9.82E-05	9.82E-05	1	87.00	39.898	1.10E-08
Feature	3.12E-04	3.12E-04	1	87.00	126.585	< 2.2e-16
Condition:Feature	8.51E-05	8.51E-05	1	87.00	34.597	7.37E-08

Table 23. Pairwise comparison of LME (Dataset3, Turkish stimuli sentences vs words)

contrast	estimate	SE	df	t.ratio	p.value
Syllables Phoneme Onset – Words Phoneme Onset	-1.24E-04	4.05E-04	87	-0.307	0.9899
Syllables Phoneme Onset – (Syllables Pho Surp/Ent)	-1.54E-03	4.05E-04	87	-3.797	0.0015
Syllables Phoneme Onset – (Words Pho Surp/Ent)	-5.03E-03	4.05E-04	87	-12.422	<.0001
Words Phoneme Onset – (Syllables Pho Surp/Ent)	-1.41E-03	4.05E-04	87	-3.489	0.0042
Words Phoneme Onset – (Words Pho Surp/Ent)	-4.91E-03	4.05E-04	87	-12.115	<.0001
(Syllables Pho Surp/Ent) – (Words Pho Surp/Ent)	-3.49E-03	4.05E-04	87	-8.626	<.0001

3. Lines 400-411. To support the statements/conclusions based on contrasting phoneme/edge

encodings in familiar vs unfamiliar uncomprehended speech, authors would need to directly compare encoding accuracy and weights across different participant groups with appropriate between-groups tests. The same holds for conclusions drawn from comparisons of phoneme/edge encoding in words vs random syllables (pseudowords) across different languages (Dutch, Turkish, Mandarin; lines 488 – 500).

We thank the reviewer for the helpful reminder regarding the need for a between-groups test for familiar versus unfamiliar uncomprehended speech. We agree that this is important, as the familiarity effect is a between-group factor. Accordingly, we conducted a new linear mixed-effects (LME) analysis including the within-subject factors *Stimuli* (Native vs. Nonnative) and *Feature* (Acoustic edge vs. Phonemes), and the between-group factor *Group* (Dutch vs. Chinese). This analysis revealed a significant three-way interaction.

Consistent with our previous analyses in Dutch participants alone, phoneme feature accuracy for the native language was higher than for the nonnative language (t ratio = 4.021, df = 81, p = 0.0031). The full set of pairwise comparisons is provided in the Supplementary Materials. For the other datasets, we did not perform between-groups comparisons, as these datasets were collected in separate experiments with different stimuli, making cross-dataset comparisons inappropriate. Nevertheless, the observed patterns are consistent across datasets: phoneme accuracy is higher for words compared to syllables, while acoustic edge accuracy shows no significant difference (Dataset 2: Dutch and Chinese; Dataset 3: Turkish). Similarly, no significant differences were observed between sentences and words for either acoustic edge or phoneme accuracy in Dataset 1 (Dutch) and Dataset 3 (Turkish).

Table 4. Between-groups LME results for Dutch and Chinese participants

	Sum Sq	Mean Sq	NumDF	DenDF	F value	Pr(>F)
Stimuli	6.94E-06	6.94E-06	1	81.00	3.83	5.38E-02
Feature	5.85E-06	5.85E-06	1	81.00	3.23	7.60E-02
Group	6.43E-06	6.43E-06	1	27.00	3.55	7.04E-02
Stimuli:Feature	6.02E-06	6.02E-06	1	81.00	3.32	7.20E-02
Stimuli:Group	4.72E-06	4.72E-06	1	81.00	2.60	1.10E-01
Feature:Group	3.86E-08	3.86E-08	1	81.00	0.02	8.84E-01
Stimuli:Feature:Group	1.14E-05	1.14E-05	1	81.00	6.30	1.41E-02

4. Lines 403-406. To support the statements/conclusions about peak latency differences, authors would need to conduct formal tests contrasting peak latencies in different conditions within the relevant participant groups (c.f. Broderick et al. 2021). Same for statements in lines 518-526 when contrasting latencies of encoding in weights for familiar vs native language in Mandarin speakers.

In our original analysis, we compared peak times for native versus non-native stimuli only at the early and mid-intervals for acoustic edges and at the late interval for phoneme features. This was based on previous findings indicating differences specifically within these windows, and we were not focused on other contrasts. However, following the reviewer’s suggestion and given that peak times were not normally distributed we conducted an ANOVA using aligned rank transformation (ART) with the `artlm()` function from the ARTool package in R. The model included the main effects of Stimuli

(Native vs. Nonnative), Group (Dutch vs. Mandarin speakers), Feature (Acoustic Edge vs. Phoneme), and Time (Early, Mid, Late).

This analysis revealed significant main effects of Stimuli, with earlier peak times for native languages ($F(1,324) = 15.23, p = 0.0001$), Feature, with earlier peak times for acoustic edges ($F(1,324) = 5.81, p = 0.0165$), and Group ($F(1,324)=3.92,p=0.0486$), with the Mandarin group showing earlier peak times overall. We also observed a significant three-way interaction among Stimuli, Feature, and Time ($F(2,324)=3.46,p=0.0327$) and among Stimuli, Group, and Time ($F(2,324)=3.66,p=0.0268$).

Pairwise comparisons of native versus non-native stimuli for each combination of Feature, Group, and Time showed only one marginally significant effect: for Mandarin speakers, phoneme features in the late interval exhibited earlier peak times for native language stimuli ($t(324) = -1.83, p = 0.068$).

We have replaced the results of our earlier analysis with the findings from this updated ART-based analysis.

Reviewer #2 (Remarks to the Author):

REVIEW

SUMMARY

This study was looking at several previously published datasets in order to examine new questions. The authors examines how different sources of predictive information modulate acoustic encoding. They specifically looked at linguistic structure (lexical knowledge) and linguistic familiarity (experience with a language). Participants were speakers of Dutch, Mandarin, and Turkish, and their MEG was measured. MEG responses to acoustic edges and phoneme boundaries were examined, and the results showed primarily that experience with a language (either by familiarity or through lexical-level information) modulates phoneme-level processing. Processing of acoustics (e.g. edges) was not modulated by experiential differences in most analyses.

MAJOR CONCERNS

I enjoyed reading this paper, and I found it added nicely to the existing literature. However, I did have some major concerns that I hope the authors could address in a revision.

First, I think the introduction needs to be expanded and clarified. The authors understandably focus on studies with MEG that use a similar continuous-in-time analysis to natural speech stimuli, but there are also several other theoretical models of speech aside from Marslen-Wilson that could help frame their rationale for a broader audience.

We thank the reviewer for their constructive feedback. We added more theoretical models of speech that also suggest an interaction across multiple levels.

These findings suggest that the phase alignment of neural signals with acoustic and linguistic features of speech is dynamically modulated through interactions across multiple levels of linguistic processing during comprehension (Altmann & Steedman, 1988; Dahan, 2010; Davis and Sohoglu, 2020; Gaskell and Marslen-Wilson, 1997; Hagoort, 2013; Luce and Pisoni, 1998; Magnuson et al., 2018; Marslen-Wilson & Welsh, 1978; Martin, 2016; Martin et al., 2017; Mattys et al., 2005; McClelland and Elman, 1986; Norris, 1994; Norris and McQueen, 2008; Pitt & McQueen, 1998; Samuel, 1997; Strauss, Harris, and Magnuson, 2007; Trueswell et al., 1994; Tyler et al., 2000; Vitevitch & Luce, 1998; Weber & Scharenborg, 2012). This modulation reflects both prior language experience and the statistical regularities learned over time (e.g., Martin & Doumas, 2017; Martin, 2016, 2020). Within this interactive framework, all subsystems engage in continuous bidirectional communication, rapidly incorporating relevant information from parallel components while simultaneously providing their own computations to other subsystems as they become available. As exposure to a nonnative language increase, linguistic categories such as phonemes become more precisely tuned, reducing reliance on raw acoustic detail because fine-grained sensory information becomes less essential. Likewise, in a native language, higher-level linguistic structure, such as words and sentences, can further refine the tuning of abstract linguistic units across layers of processing.

Similarly, in the Discussion, the authors say:

“Our analyses suggest that statistical information can be obtained either from the mental lexicon, in the case of native language comprehension, or from experience with a spoken language, in the absence of language comprehension. Even short-term repeated exposure to an uncomprehended spoken language, as in the experimental setups for the Dutch and Mandarin Chinese languages, can increase the tracking of phoneme features by the brain; yet the time dynamics of phoneme feature-tracking show distinct profiles in a native versus nonnative language.”

I would love to hear a more nuanced discussion of this from the authors. They do a good job of explaining what they found in their results, but I would love to hear more of these types of bigger picture theoretical implications. Some literature that to me seems particularly relevant and might be worth looking into is the lexicalization of nonwords—where exposure to the words can cause them to sufficiently lexicalize and behave in many ways like words, even without semantic meaning. The authors mention exposure to a language in terms of statistical regularities, but I wonder if this nonword lexicalization could be another source of predictive information that could affect acoustic-phonemic processing.

We appreciate the suggestions to extend our discussion related to nonword lexicalization. We added below paragraph to Exposure to an uncomprehended language enhances phoneme feature tracking section of the discussion.

Another possible explanation for the delayed phoneme peak in the nonnative language is competition between newly lexicalized nonnative word forms (without semantic mappings) and existing native words. Prior research shows that repeated exposure to nonwords (in this case words in an uncomprehended language) can lead to their lexicalization without meaning (Gaskell & Dumay, 2003; Kapnoula et al., 2015). Because these newly formed lexical entries have weaker memory traces, they require more time to process (McClelland & Elman, 1986; Norris, 1994; Luce & Pisoni, 1998).

Finally, in the Methods and throughout the Results, I would like to hear more details about the statistical models, and I would appreciate it if the authors could report their F-ratios and t-statistics in the typical convention, with degrees of freedom included:

$F(df_{\text{between}}, df_{\text{within}}) = F\text{-ratio}, p = p\text{-value}$

$t(df) = t\text{-value}, p = p\text{-value}$

We thank the reviewer for their valuable feedback regarding the reporting of statistical details. In response to your comment, we have now expanded the statistical reporting throughout both the Methods and Results sections. While we had previously included F-ratios and t-statistics with their associated degrees of freedom in the appendix, we recognize the importance of having this information readily accessible in the main text. Accordingly, we have now incorporated these statistics directly into the Results section, following the standard conventions you specified.

Also, as a nonexpert on acoustic edges, I did not fully understand what this really refers to in terms of the features of the acoustic signal. Could the authors please clarify early on in the manuscript and maybe further elucidate with an updated visualization in Figure 1A?

We thank the reviewer for this helpful suggestion. To address this point, we have added a paragraph in the Introduction that more clearly defines “acoustic edges”. We hope that this explains the Figure 1A better.

Linguistic features of speech can be correlated with the acoustic features, and for example the neural tracking of phonemic features can also be explained by the tracking of rapid changes in acoustic envelope (e.g., acoustic edges; Daube et al., 2019; Doelling et al. 2014; Oganian et al. 2023). The acoustic envelope captures slow amplitude changes in speech signals, with its temporal patterns

reflecting the timescales of syllables and words. Phoneme onsets correspond to acoustic onsets, termed “acoustic edges”, which are derived from half-wave rectified derivatives of the acoustic envelope (Brodbeck et al., 2018; Daube et al., 2019). To separate neural responses driven by acoustic versus linguistic information at the phoneme level, we incorporated acoustic edge features (an 8 logarithmically spaced frequency band acoustic onset spectrogram, which models sudden changes in the gammatone spectrogram of the audio signal using an acoustic edge detection model) alongside an auditory spectrogram with 8-band that represents the acoustic envelope (Fig 1A, See Methods).

Methods:

Acoustic Features: Acoustic features were generated using the Eelbrain toolbox (Brodbeck, 2023). This included an 8-band gammatone spectrogram and an 8-band acoustic onset spectrogram, which models sudden changes in the gammatone spectrogram of the audio signal using an acoustic edge detection model (Fishbach et al., 2001). Both spectrograms covered frequencies from 20 to 5000 Hz in equivalent rectangular bandwidth (ERB) space.

MINOR CONCERNS

In Introduction, the authors may also consider broadening and adding some discussion of why acoustic features are less encoded with higher proficiency, particularly in contrast with the idea of perceptual tuning?

We thank the reviewer for bringing these to our attention. We believe these results are not contradictory to perceptual tuning. In native-language processing, more relevant information such as abstract linguistic units like phonemes is fine-tuned, while detailed and potentially noisy acoustic information is suppressed relative to a nonnative language. Consequently, acoustic details may play a less critical role in a native language than in a nonnative one. We have added the paragraph below to clarify this point.

As exposure to a nonnative language increase, linguistic categories such as phonemes become more precisely tuned, reducing reliance on raw acoustic detail because fine-grained sensory information becomes less essential. Likewise, in a native language, higher-level linguistic structure, such as words and sentences, can further refine the tuning of abstract linguistic units across layers of processing (Kraljic and Samuel, 2005; Norris, McQueen, and Cutler, 2003; Davis et al., 2005).

Line 168, “more words in a sentence”—Was this also a statistically significant difference? Please report in that same area. Moreover, was the number of phonemes or acoustic edges per sentence different?

We did not perform a statistical analysis comparing the number of words per sentence, as each sentence in the Dutch stimuli contained exactly ten words, while each sentence in the Turkish stimuli contained four words. To clarify this point, we added the term *exactly* in the previous sentence. Instead, we compared the datasets under the assumption that the observed differences in TRF weight patterns for phoneme features might arise from variations in the contextual information of the stimuli. The lower surprisal values for words in the Dutch dataset suggest that words were more predictable given the preceding context. Conversely, the number of phonemes or acoustic edges is unlikely to account for the TRF weight differences, as the encoding model already captures the average neural response to phonemes and acoustic edges.

Additionally, we realized that this analysis was mentioned at the beginning of the Results section without a clear explanation of its purpose. Therefore, we revised the section as shown below and moved this paragraph to the end of the Results section for better coherence.

Although the improvement in prediction accuracy due to acoustic edges and phoneme features did not differ significantly between the sentence and word conditions in either dataset, the TRF weights

exhibited distinct patterns across datasets. In the Dutch dataset, phoneme feature weights for sentences were greater than those for words throughout the entire time interval. In contrast, the Turkish dataset showed the opposite pattern during the early time window (–30 to 140 ms), but phoneme feature weights were again higher for sentences than for words in the later time window (370 to 570 ms). This difference in TRF weights may stem from variations in the contextual information of the stimuli or in the methods used to generate them. Each sentence had exactly 10 words in Dutch stimuli and 4 words in Turkish stimuli (see Methods). We compared the predictability of each word in sentences in Dutch and Turkish stimuli by calculating the surprisal values of each word using a GPT2 model (Dutch: Havinga, 2024; Turkish: Kesgin et al, 2024). Words in Dutch sentences had a lower surprisal value (more predictable) than words in Turkish stimuli ($p < 0.0001$, mean surprisal Dutch words=13.16, std=5.46; mean surprisal Turkish words=14.33, std=3.79) possibly due to a longer context – more words in a sentence – in Dutch stimuli). In the Turkish dataset, the words were isochronous, meaning that the onset of each word was perfectly predictable. In contrast, the Dutch dataset contained naturally spoken words with variable onset times and durations. We discussed how these differences might have contributed to the divergent results in the Discussion section.

Line 175, This is picky but should LLM be abbreviated as LME, (to me, LLM = large language Model)

We thank the reviewer for pointing this out. It was written as LLM in one place by mistake instead of LMM but now we revised them all as LME.

Line 203, 700 ms seems like a short epoch, why was this chosen as such?

The time window was set to 700 ms, as we did not expect neural responses to acoustic edges or phonemes to extend beyond this duration. Previous studies employing the TRF method for encoding acoustic edges and phoneme features used time windows ranging from 500 ms to 1000 ms but reported no significant effects beyond 700 ms (Di Liberto et al., 2015; Brodbeck et al., 2018; Daube et al., 2019; Donhauser & Baillet, 2020; Brodbeck et al., 2022; Tezcan et al., 2023; Gillis et al., 2023).

We also added this to the methods section in the manuscript.

Line 204, Please add more details about cluster-based permutation?

We revised the methods as below.

To investigate the effect of condition on TRF weights, average of the absolute values of weights over sources for each time lag were compared by using cluster permutation test across time with 30000 permutations (Oostenveld, Fries, Maris, & Schoffelen, 2011) implemented on mne- python (version 0.23.1).

Line 235, Authors can just use LME again since they already introduced it

Corrected.

Line 246 and elsewhere, If you are reporting LME output, why do you have F ratios? Should these be beta coefficients? What package was used to run the LMEs? Was it the LME4 package in R?

We used the lme4 package in R for our linear mixed-effects models. We chose to report F-ratios from Type III ANOVA tests (using the lmerTest package's anova() function on the fitted models) rather than individual beta coefficients, as this approach provides a clearer overview of main effects and interactions.

While we had initially included R package information in the Methods section, we have now also added it to the beginning of the Results section as below to ensure clarity about our analytical approach.

We used lme4 package to fit LMEs and reported the output of the lmerTest package's anova() function for the fitted model to obtain Type III F-statistics with Satterthwaite-approximated degrees of freedom and demonstrate the main and interaction effects.

Line 303, typo, "listening to Mandarin Chinese stimuli"

Corrected

Line 331 & 403, typo, "uncomprehended"

Corrected

Line 412-3, typo, the period should go after the citations

Corrected

Reviewer #3 (Remarks to the Author):

The study by Tezcan et al. analyzes three datasets to examine how both linguistic structure and experience-based familiarity shape neural tracking of acoustic and phonemic features. Across datasets, the authors consistently find that phonemic tracking becomes stronger as linguistic processing deepens—whether through the construction of larger linguistic units, greater language proficiency, or increased familiarity with the language.

Overall, the paper offers a comprehensive view of how acoustic and phonemic responses are modulated by linguistic structure and familiarity. I enjoyed the abstract and the opening of the introduction; both adopt a broad, insightful perspective and clearly summarize the compelling main results. My main concerns relate to the organization and presentation of the results, with a view to improving readability and helping readers extract the key findings easier.

1. The inconsistent results between reconstruction accuracy and TRF weights. First, the reconstruction accuracy is not formally defined in this manuscript. If I understand correctly, the reconstruction accuracy represents the R square value between the predicted and measured MEG signal? If so, the 'reconstruction' should be called as 'predictive/prediction' as reconstruction usually refers to reconstructing speech features (e.g., the envelope) from neural signals.

We thank the reviewer for their feedback. We added the model accuracy description in the Methods section as below and replaced "model reconstruction accuracy" with "model accuracy".

Model accuracy was calculated as the proportion of variance in the neural signal explained by the model.

The reconstruction accuracy generally shows a consistent pattern as TRF weights, which means if the neural response to a feature is larger, then the accuracy in predicting the response using this feature would also be larger. As summarized in Fig. 6, such inconsistency often appears in the feature of acoustic edges. Drawing a conclusion given such inconsistency may not fully convince readers. I suggested the authors offer a plausible explanation for the inconsistency or reconsider this claim.

We thank the reviewer for highlighting this point. We realized that we had not provided sufficient explanation of these measures in the Introduction and had only described their meaning in the Discussion. Model accuracy improvements and model weight differences capture distinct aspects of the analysis and, therefore, are not necessarily expected to be consistent with each other. We have now added a clarification in the Introduction explaining what each of these measures reflects, as shown below.

We compared the improvement in model accuracy attributed to the inclusion of acoustic and phoneme features, defined as the increase in the proportion of neural signal variance explained by the model when these features were added. This measure reflects how well the real neural signal aligns with the model's predicted signal in terms of both phase and power.

In addition, we compared the model weights, the filters learned by the model that represent the neural responses evoked by specific features. This measure enables us to assess differences in the amplitude of neural responses contributing to the predicted signal.

We also added in our discussion below paragraph:

The differential patterns we observe where word-syllable contrasts affect both accuracy and weights, while sentence-word contrasts affect only weights likely reflect distinct underlying neural mechanisms. When processing random syllables versus words, the brain might be recruiting distinct cortical circuits with different oscillatory signatures, resulting in differences in both phase alignment (captured by accuracy) and power modulation (reflected in weights). In contrast, words presented in isolation versus within sentences might be utilizing the same network, but with altered temporal dynamics. The sentence structure might modulate how this network processes word-level information adjusting the timing and strength of responses rather than recruiting different circuits.

Additionally, from my perspective, the main results of this study are phoneme encoding, which are overall consistent in terms of prediction accuracy and TRF weights, and are interesting enough. I am uncertain if the claim about acoustic responses enhances the current study a lot or just adds confusion.

We believe that the difference in acoustic edge tracking, which appears to be modulated by language comprehension rather than by linguistic structure, makes an important contribution to our findings. This pattern suggests that linguistic structure primarily influences abstract linguistic units, such as phonemes, while acoustic information may be suppressed due to the lexicalization of sounds, or predictability. We hope that our response to the next comment will help clarify this point further.

2. Also, how the basic acoustic response is modulated by structural building or by language familiarity has been studied in many relevant literatures, for either natural speech (Decruy et al., 2019, 2020; Reetzke et al., 2021; Zou et al., 2019) and isochronous speech (Chen et al., 2020), which consistently show that basic acoustic responses decrease with higher language proficiency. These findings are not consistent with the results of the current study. More detailed discussion is need.

We thank the reviewer for this thoughtful comment and the opportunity to clarify this point. We believe there may have been a misunderstanding. Our results also indicate that acoustic responses decrease with higher language proficiency when comparing levels of language familiarity. As shown in Figure 5, model weights for acoustic edges are lower for the native language during the early time interval. We did not observe a difference in acoustic edge tracking between sentences and words, likely because both conditions involve comprehensible speech for native listeners. This pattern suggests that language structure does not modulate acoustic edge tracking directly, but rather influences more abstract linguistic representations, such as phonemes.

Regarding the suggested studies, we did not initially include them in our discussion because they did not examine acoustic and linguistic feature tracking within a multivariate framework. As noted in the Introduction, linguistic and acoustic features of speech are often correlated. For example, neural tracking of phonemic features may partly reflect sensitivity to rapid changes in the acoustic envelope (e.g., acoustic edges; Daube et al., 2019; Doelling et al., 2014; Oganian et al., 2023). For this reason, we primarily compared our findings with studies that employed a similar multivariate approach, with which our results were consistent. However, in response to the reviewer's suggestion, we have now incorporated references to studies related to language experience and attention that used speech envelope measures in the Introduction.

“(Ahissar et al., 2001; Nourski et al., 2009; Hertrich et al., 2012; Zou et al, 2019), attention (Lakatos et al., 2008; Schroeder and Lakatos, 2009; Gomez-Ramirez et al., 2011, Teoh et al., 2022), language experience (Reetzke et al,2021; Chen et al, 2020; Brodbeck et al., 2024, Pérez-Navarro et al, 2024)”

3. Some comparisons of TRF weights appear a bit problematic. For example, Figures 2C and 4D (which should be labeled 4E) seem to represent a baseline difference between the two waveforms, resulting in significant differences across all time lags, even before zero. Authors should verify whether these significant differences reflect meaningful effects or arise from baseline differences, as the power of TRF weights could be affected by the noise levels and samples for the average.

We thank the reviewer for bringing this to our attention. We have revised the manuscript to correct the figure label, which is now properly indicated as “Fig. 4E.”.

Regarding the subsequent comment, this effect cannot arise from a baseline difference. The TRF estimates time-invariant responses for each feature directly from the data; these responses are not baselined and are computed over the same time interval (–50 to 700 ms). In addition, both the features and the segmented neural data were normalized prior to model training by subtracting the mean and dividing by the standard deviation.

Responses appearing before 0 ms are plausible due to coarticulation, whereby information about a phoneme’s identity may already be present in the acoustic signal prior to its conventional onset (Beddor et al., 2013; Salverda et al., 2003; Brodbeck et al., 2020). This effect is particularly pronounced for the isochronous Turkish stimuli, in which word lengths were equalized by temporal stretching or compression, resulting in no silence between words or syllables. In contrast, the Dutch and Mandarin stimuli were made rhythmic by inserting silent intervals between words.

4. Does Figure 5A compare Chinese-speaking participants listening to Mandarin versus Dutch speech? If so, differing speech materials complicate interpretation. A more appropriate comparison might involve participants with different language familiarity listening to identical materials—for example, comparing Chinese and Dutch participants both listening to Dutch.

We appreciate the reviewer’s insightful comment. Indeed, Figure 5A compares Chinese-speaking participants listening to Mandarin versus Dutch speech, while Figure 5B compares Mandarin-speaking participants listening to Mandarin versus Dutch speech.

Comparing Chinese-speaking and Dutch-speaking participants listening to the same stimuli would not allow us to examine language familiarity with a nonnative language, since the stimuli would correspond to the native language of only one of the participant groups. Moreover, such a comparison could introduce confounding factors related to cross-group differences in neural responses. In particular, differences in model accuracy or model weights might reflect variations in signal-to-noise ratio between participant groups rather than effects of language familiarity per se. Importantly, acoustic and phonemic feature variability has already been modeled in our analysis, and the observed differences in neural responses reflect the contrast between native and nonnative language processing for both groups. However, the two groups differ in their degree of familiarity with the nonnative language, which is precisely what our comparison aims to capture. For this reason, we chose to conduct the analysis as presented.

5. As this study is conducted based on multiple datasets, including multiple languages and multiple structures, it gives a comprehensive view of how acoustic and phonemic encoding are affected by structure building and language familiarity (which I admire and appreciate). However, it also brings the difficulty of figuring out exactly what stimuli were used in each dataset and what the corresponding tasks were. I suggest that the authors reorganize Figure 1 or include a separate figure that illustrates the structure of the stimuli, the task, and the participants, so that readers don’t need to refer to the Methods section or references to find this information.

6. *The discussion is currently challenging to navigate. To improve readability, the authors may consider structuring it with subheadings.*

We thank the reviewer for this constructive suggestion. We have revised the Discussion section to improve readability by adding subheadings that highlight the main findings of each subsection.

References

- Chen, Y., Jin, P., & Ding, N. (2020). The influence of linguistic information on cortical tracking of words. *Neuropsychologia*, 148, 107640. <https://doi.org/10.1016/j.neuropsychologia.2020.107640>
- Decruy, L., Vanthornhout, J., & Francart, T. (2019). Evidence for enhanced neural tracking of the speech envelope underlying age-related speech-in-noise difficulties. *Journal of Neurophysiology*, 122(2), 601–615. <https://doi.org/10.1152/jn.00687.2018>
- Decruy, L., Vanthornhout, J., & Francart, T. (2020). Hearing impairment is associated with enhanced neural tracking of the speech envelope. *Hearing Research*, 393, 107961. <https://doi.org/10.1016/j.heares.2020.107961>
- Reetzke, R., Gnanateja, G. N., & Chandrasekaran, B. (2021). Neural tracking of the speech envelope is differentially modulated by attention and language experience. *Brain and Language*, 213, 104891. <https://doi.org/10.1016/j.bandl.2020.104891>
- Zou, J., Feng, J., Xu, T., Jin, P., Luo, C., Zhang, J., Pan, X., Chen, F., Zheng, J., & Ding, N. (2019). Auditory and language contributions to neural encoding of speech features in noisy environments. *NeuroImage*, 192, 66–75. <https://doi.org/10.1016/j.neuroimage.2019.02.047>